# Multi-omics identify falling LRRC15 as a COVID-19 severity marker and persistent pro-thrombotic signals in convalescence

Jack S. Gisby [1,3], Norzawani B. Buang [1,3], Artemis Papadaki[1], Candice L. Clarke[1,2], Talat H. Malik[1], Nicholas Medjeral-Thomas[1,2], Damiola Pinheiro[1], Paige M. Mortimer[1], Shanice Lewis[1], Eleanor Sandhu[1,2], Stephen P. McAdoo[1,2], Maria F. Prendecki [1,2], Michelle Willicombe[1,2], Matthew C. Pickering [1], Marina Botto [1], David C. Thomas [1,2,4] & James E. Peters [1,4]

Patients with end-stage kidney disease (ESKD) are at high risk of severe COVID-19. Here, we perform longitudinal blood sampling of ESKD haemodialysis patients with COVID-19, collecting samples pre-infection, serially during infection, and after clinical recovery. Using plasma proteomics, and RNA-sequencing and flow cytometry of immune cells, we identify transcriptomic and proteomic signatures of COVID-19 severity, and find distinct temporal molecular profiles in patients with severe disease. Supervised learning reveals that the plasma proteome is a superior indicator of clinical severity than the PBMC transcriptome. We show that a decreasing trajectory of plasma LRRC15, a proposed co-receptor for SARS-CoV-2, is associated with a more severe clinical course. We observe that two months after the acute infection, patients still display dysregulated gene expression related to vascular, platelet and coagulation pathways, including *PF4* (platelet factor 4), which may explain the prolonged thrombotic risk following COVID-19.

COVID-19, caused by the SARS-CoV-2 virus, is a highly heterogeneous disease. In most individuals, it is a mild, self-limiting illness, but some individuals develop severe disease, typically manifesting as respiratory failure with marked systemic inflammation and immunopathology. Multiple studies have described immunological[1,2], transcriptomic[3–11], and proteomic[12–20] correlates of severe disease. The importance of an aberrant host immune response in tissue injury in severe COVID-19 is supported by the efficacy of anti-inflammatory treatments. These include glucocorticoids[21], monoclonal antibodies blocking the interleukin-6 receptor[22,23], and the Janus kinase (JAK) inhibitor baricitinib[24]. A wide range of additional therapies directed at specific elements of the inflammatory response has been developed for immuno-inflammatory diseases and present potential repurposing opportunities for the treatment of severe COVID-19. Understanding the molecular basis for severe COVID-19 is important for the rational selection of such therapies.

Risk factors for severe COVID-19 include age, male sex, and the presence of comorbidities such as chronic kidney disease (CKD). In CKD, the risk of severe COVID-19 is proportional to the degree of renal impairment[25]. End-stage kidney disease (ESKD) confers particularly high risk, with a population-based study estimating a hazard ratio for death of 3.69[25] and a European registry study reporting 23.9% 28-day mortality in dialysis patients with COVID-19[26]. In part, this is because ESKD patients are enriched for other risk factors for severe COVID-19, including cardiometabolic disease. However, even after adjustment for these, ESKD remains independently associated with the risk of severe

[1]Centre for Inflammatory Disease, Dept of Immunology and Inflammation, Imperial College London, London, UK. [2]Renal and Transplant Centre, Hammersmith Hospital, Imperial College Healthcare NHS Trust, London, UK. [3]These authors contributed equally: Jack S. Gisby, Norzawani B. Buang. [4]These authors jointly supervised this work: David C. Thomas, James E. Peters. ✉ e-mail: david.thomas1@imperial.ac.uk; j.peters@imperial.ac.uk

COVID-19. In addition, ESKD patients display impaired vaccine responses[27,28], and those on in-centre haemodialysis cannot shield effectively during lockdowns as they need to access dialysis facilities regularly.

Here, we investigated the host response to SARS-CoV-2 in ESKD patients on haemodialysis since study of such an at-risk group should enhance the probability of identifying severity signals and might also point to either an exaggerated or even distinct immunological response to the virus. Moreover, ESKD patients receiving haemodialysis present an opportunity for serial blood sampling of both outpatients and inpatients with COVID-19, since patients must attend medical facilities for regular dialysis regardless of COVID-19 severity. This enabled us to perform longitudinal analysis and avoid the selection bias that affects studies limited solely to hospitalised patients.

The host response to SARS-CoV-2 is orchestrated by a complex network of cells and mediators, including circulating proteins such as cytokines and soluble receptors. Soluble proteins play key roles in multiple biological processes, including signalling, host defence and repair, and are potential biomarkers and therapeutic targets. We therefore hypothesised that a comprehensive analysis of both circulating proteins and immune cells should yield valuable and complementary insights into the pathobiology of COVID-19. To this end, we used the aptamer-based SomaScan platform that provides broad coverage of the plasma proteome (6323 proteins), combined with RNA-sequencing and flow cytometry of peripheral blood mononuclear cells (PBMCs). We integrated these data to provide a comprehensive view of the COVID-19 multi-omic landscape, enabling us to link transcriptomic and cellular changes with circulating proteins. Supervised learning identified plasma levels of the LRRC15 protein, a recently proposed alternative receptor for SARS-CoV-2, as a marker of disease severity. By comparing pre-infection samples to samples collected from the same individuals during COVID-19 and after clinical recovery, we revealed persistent upregulation of gene expression signatures related to vascular and clotting pathways several months after infection. These findings elucidate the biological underpinnings of the prolonged pro-thrombotic state associated with COVID-19.

## Results

### Features of patient cohorts

We recruited two cohorts of ESKD patients on haemodialysis presenting with COVID-19 (Fig. 1a). The Wave 1 cohort consisted of 53 patients recruited during the initial phase of the COVID-19 pandemic (April-May 2020) (Supplementary Table 1). Serial blood sampling was carried out where feasible (Fig. 1b), given the pressure on hospital services and the effects of national lockdown. We assessed disease severity using a WHO four-level ordinal score, categorising it into mild, moderate, severe, and critical. Of the 53 patients, 25 had a peak illness severity score of severe or critical (hereafter severe/critical) and 28 mild or moderate (mild/moderate). Nine died. The majority of patients were of non-European ancestry. Further clinical and demographic details are provided in Supplementary Table 1. We also contemporaneously recruited 59 non-infected haemodialysis patients to provide a control group, selected to mirror the age, sex and ethnicity distribution of the COVID-19 cases (Supplementary Fig. 1a–c).

The Wave 2 cohort consisted of 17 ESKD patients with COVID-19, infected during the resurgence of cases in January-March 2021 (Supplementary Table 2). All had been recruited as part of the COVID-19 negative control group during Wave 1, thereby providing a pre-infection sample collected 8–9 months earlier. For the Wave 2 cohort, we systematically acquired serial samples for all patients at regular intervals (every 2–3 days over the course of the acute illness) (Fig. 1c). 9 patients had a peak illness severity of severe/critical (of whom 4 died), and 8 mild/moderate. For 12 of these patients, we acquired convalescent samples approximately two months following infection.

### The effect of COVID-19 on the PBMC transcriptome and plasma proteome in ESKD patients

We performed transcriptomic profiling using RNA-seq of PBMCs. Principal components analysis (PCA) revealed a clear effect of COVID-19 in both Wave 1 (COVID-19 positive and negative patient samples) and Wave 2 (pre-infection and subsequent COVID-19 positive samples from the same individuals) (Fig. 2a). In the Wave 1 cohort, differential gene expression analysis between COVID-19 positive ($n = 179$ samples from 51 patients) and negative samples ($n = 55$) (using linear mixed models (LMM) to account for repeated samples from the same individuals) identified 3026 significantly up-regulated and 3329 down-regulated genes (1% false discovery rate, FDR) (Supplementary Data 1a). Sensitivity analyses exploring the effects of including additional clinical covariates (underlying cause of ESKD, diabetes, and time since first commencing haemodialysis) in the model did not materially impact the results (Supplementary Material, Supplementary Fig. 2). For the Wave 2 cohort, where we compared COVID-19 positive samples ($n = 90$ samples from 17 individuals) with pre-infection samples from these same individuals, we identified 2871 up-regulated and 3325 down-regulated genes (1% FDR, LMM) (Supplementary Data 1a). These findings demonstrate widespread transcriptomic changes associated with COVID-19. 3468 genes were significantly differentially expressed (1% FDR) in both the Wave 1 and 2 cohorts. However, this approach of intersecting lists of significant features based on a hard statistical threshold will underestimate commonality between two datasets[29]. To provide a measure of consistency that is not dependent on the significance threshold, we compared the estimated effect size (log2 fold change) between COVID-19 positive and negative samples for each gene in the Wave 1 and Wave 2 cohorts. These were highly concordant (Pearson's $r$ 0.80) (Supplementary Fig. 3a), despite differences in the prevalent SARS-CoV-2 variant and developments in medical management (8 of 17 patients in the Wave 2 cohort received glucocorticoids). To identify the genes that were consistently differentially expressed across both cohorts, we used robust rank aggregation (RRA) (Supplementary Data 1a, Supplementary Fig. 4).

To gain insight into the biological pathways underlying these changes, we used Gene Set Variation Analysis (GSVA)[30] to compare COVID-19 positive and negative ESKD samples (Supplementary Data 1b). Enriched pathways included those related to cell cycle (e.g., 'Polo-like kinase mediated events', which are involved in the cellular response to DNA damage) and host defence (e.g., 'Complement cascade', 'Fc-gamma receptor-dependent phagocytosis', and 'Parasite infection') (Supplementary Fig. 5). This analysis also highlighted leucocyte-endothelial interactions ('Cell surface interactions at the vascular wall', which included *SELL* and *CEACAM-1, -3, -6* and *-8* genes). Examples of marked changes in gene expression between pre-infection and acute infection in the Wave 2 cohort included components of 'Immunoregulatory interactions between a lymphoid and a non-lymphoid cell' pathway term (e.g., *SIGLEC1*, *SIGLEC9*, *SELL*, all increased) and 'Development and heterogeneity of the ILC family' (e.g., *IFNG*, *GATA3*, *RORA*, all decreased) (Fig. 2b).

We next assessed the circulating proteome, measuring 6323 proteins using the SomaScan platform (Supplementary Data 1c). PCA showed clear differences between COVID-19 positive and negative samples (Fig. 2a). We identified 1273 differentially abundant proteins between COVID-19 positive and negative samples in Wave 1 (86 samples from 37 COVID-19 positive ESKD patients versus 53 non-infected ESKD patient samples, LMM) (Supplementary Data 1d, Supplementary Fig. 6). In Wave 2, comparison of COVID-19 positive samples ($n = 102$ samples from 17 patients) with pre-infection samples from the same individuals identified 5265 differentially abundant proteins. The effect sizes were generally concordant between the cohorts (Pearson's $r$ 0.57) (Supplementary Fig. 3b), and 730 proteins were significantly differentially abundant (1% FDR) in both the Wave 1 and 2 datasets. As for our transcriptomic analysis, we used RRA to rank the consistency of

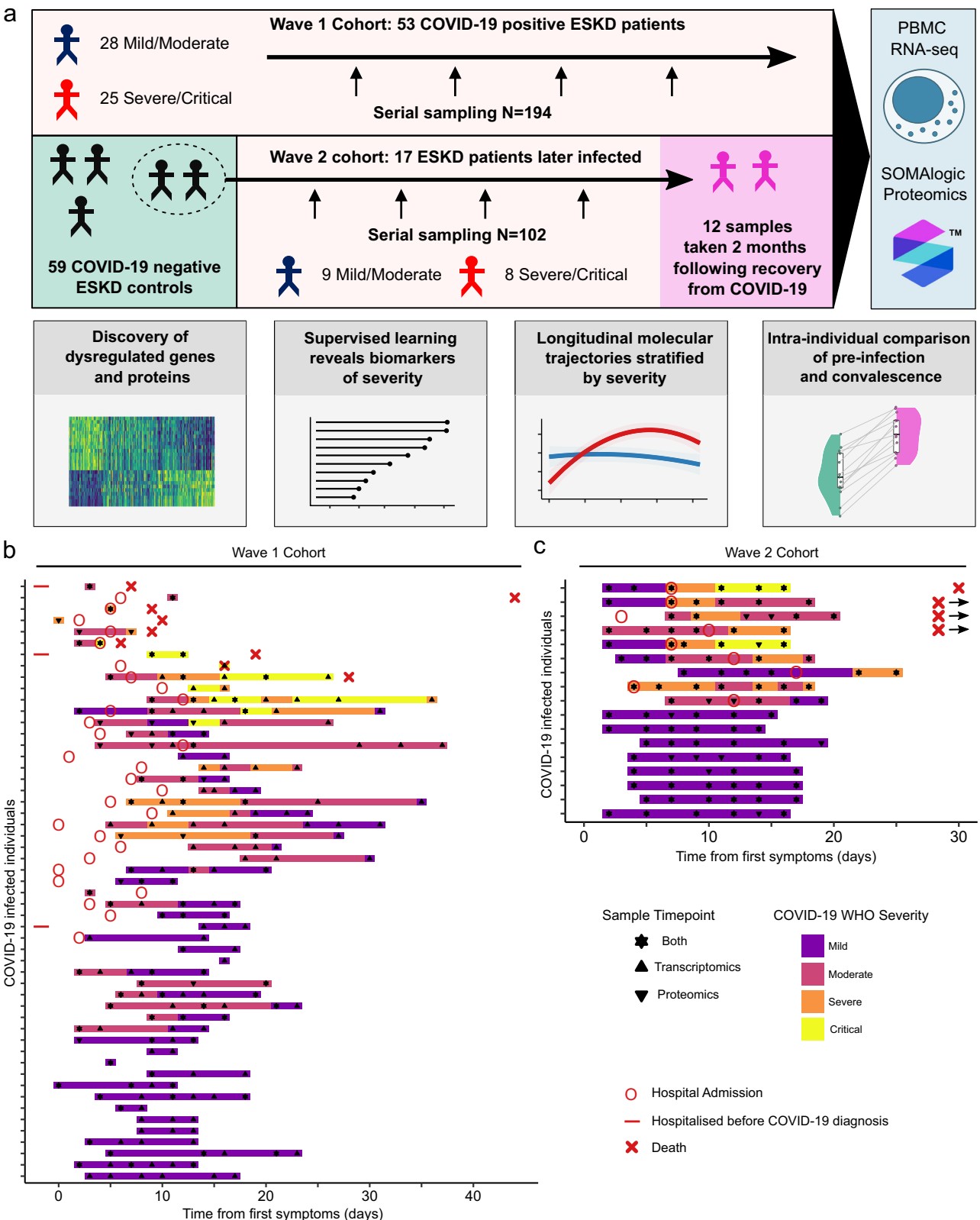

**Fig. 1 | Study design and cohort summary. a** Graphical summary of the patient cohorts, sampling, and major analyses. Wave 1 patients were recruited in Spring 2020. 17 of the COVID-19 negative ESKD patients recruited as a controls in Wave 1 were recruited again as COVID-19 positive cases in Wave 2 (2021). For 12/13 survivors in Wave 2, we obtained a convalescent sample approximately 2 months following recovery. Thus, for Wave 2, we had paired pre-infection, acute infection and post-infection samples from the same individuals. **b, c** For each cohort, the timing of the serial blood sampling is shown by triangles and the temporal COVID-19 severity by coloured bars. Three patients were hospitalised prior to COVID-19 diagnosis in the Wave 1 cohort. Three of the four patients in the Wave 2 cohort with fatal outcomes died >30 days from first positive swab.

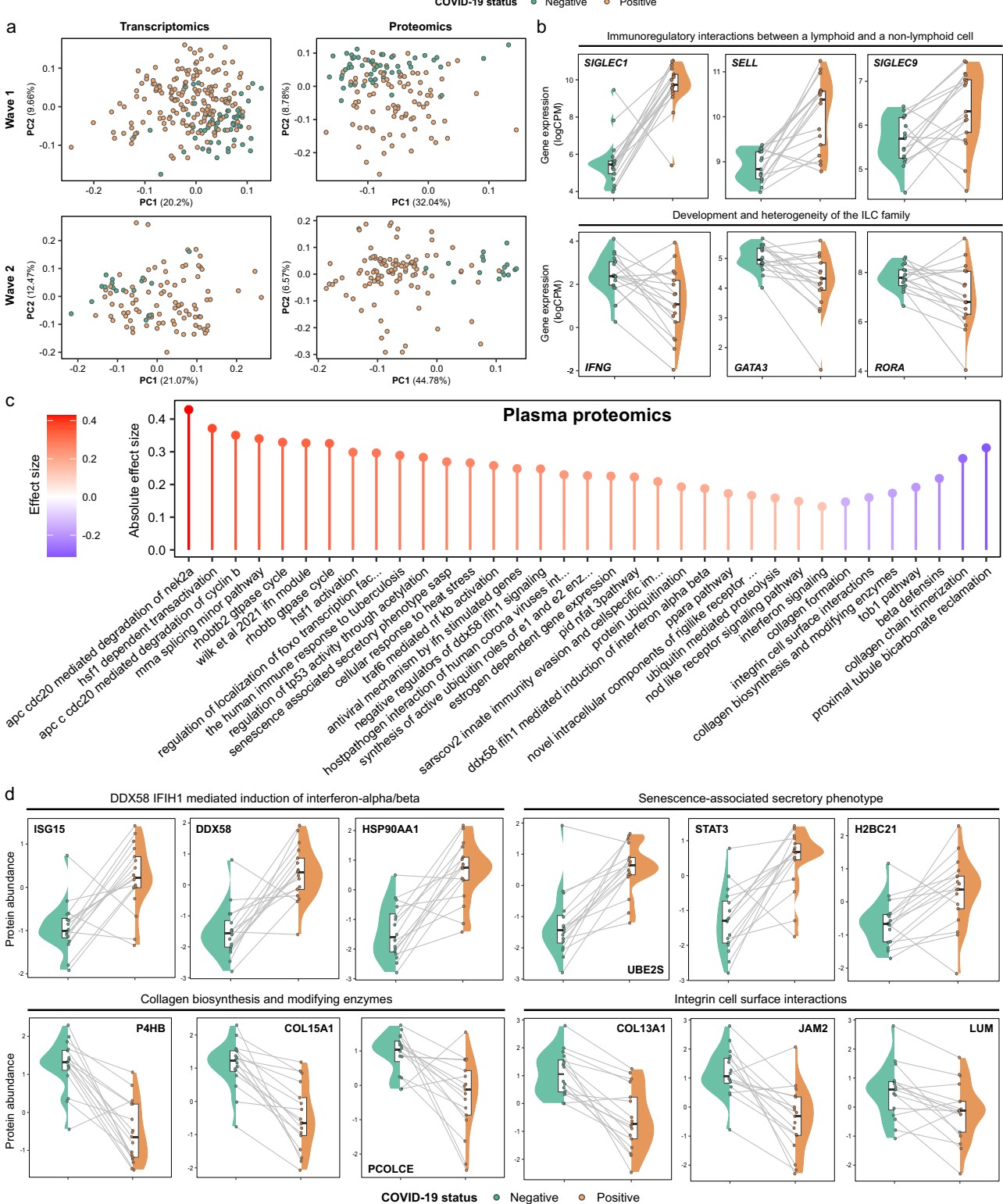

**Fig. 2 | Signatures of COVID-19 in ESKD. a** PCA of the PBMC transcriptome (left) and plasma proteome (right). Each point represents a sample and is coloured by COVID-19 status. **b** Paired violin plots showing intra-individual comparisons of pre-infection and most severe sample (Wave 2 cohort; $n = 34$ samples from 17 individuals) during COVID-19 for selected genes. Grey lines link each individual's pre-infection and infection samples; these samples are represented by points and coloured by COVID-19 status. Shaded areas indicate kernel density estimates. For boxplots, centre = median, upper bound = upper quartile, lower bound = lower quartile. All genes shown were significantly differentially expressed (1% FDR) in

both cohorts. Genes are grouped by membership to pathways that were significantly enriched (1% FDR) in GSVA. **c** The 30 protein pathway enrichment terms with the greatest RRA scores (indicating consistent dysregulation in both the Wave 1 and Wave 2 proteomic datasets), ordered by effect size. All pathway terms shown were significantly enriched in the individual cohort analyses (1% FDR). Red= up-regulated in COVID-19 positive versus negative; blue= down-regulated. **d** As for **b**, but displaying selected plasma proteins (significant at 1% FDR) ($n = 32$ samples from 16 individuals).

differentially abundant proteins across the cohorts (Supplementary Data 1d).

Enrichment analysis revealed upregulation of pathways, including 'DDX58/IFIH1 mediated induction of interferon-alpha/beta', 'Wilk et al., 2021 IFN module'[31], 'Host-pathogen interaction of human coronaviruses interferon induction' and 'SARS-CoV-2 innate immunity evasion and cell-specific immune response', reflecting host anti-viral responses and providing validation of our analysis (Fig. 2c, Supplementary Data 1e). Highly up-regulated proteins within these pathways included STAT1; DDX58 and ISG15, both crucial to the IFN-mediated antiviral response in COVID-19[32]; IFITM3, which is up-regulated in lung epithelial cells during early SARS-CoV-2 infection[33]; and the chemokines CXCL11, CXCL1, CXCL6, CXCL5 and CXCL10. Another significantly up-regulated pathway was 'Senescence-associated secretory phenotype', which included up-regulated ubiquitin-conjugating enzymes (UBE2S, UBE2E1), histones (H2BC21, H2BU1) and STAT3 (Fig. 2d). Down-regulated pathways included 'Integrin cell surface interactions' and 'Collagen biosynthesis and modifying enzymes' which contained collagen proteins (e.g., COL11A2, COL13A1, COL15A1) and related enzymes (e.g., P4HB, PCOLCE) (Fig. 2d).

## Transcriptomic and proteomic changes associated with COVID-19 severity

In both cohorts, the PCA of the PBMC transcriptomics revealed differences according to both severity at time of sampling and overall clinical course (defined by peak severity score) (Fig. 3a). There was a gradient of severity reflected in the molecular phenotype. We next assessed molecular features associated with severity at time of blood sampling, encoded as an ordinal variable. We identified 3522 genes that were significantly associated with contemporaneous severity in the Wave 1 cohort and 657 genes in the Wave 2 cohort (LMM, 1% FDR, Supplementary Data 1f, Supplementary Fig. 7). 363 genes were significantly associated in both cohorts. We then applied GSVA to identify pathways and used RRA to combine results from each cohort (Supplementary Data 1g).

The up-regulated transcriptomic pathways in more severe disease included those involved in oxidative stress ('Glutathione metabolism', 'Detoxification of reactive oxygen species'), 'Transcriptional regulation of granulopoiesis', pathways containing numerous histone-encoding genes ('HDACs deacetylate histones', 'Diseases of programmed cell death', 'RHO GTPases activate PKNs') and 'Complement and coagulation cascades' (Fig. 3b, c, Supplementary Data 1g). Down-regulated pathway terms included 'TCRA pathway', 'Pathogenesis of SARS-CoV-2 mediated by nsp9-nsp10 complex', 'TP53 activity', and 'PD1 signalling', suggesting T cell activation in more severe COVID-19 (Fig. 3b, c, Supplementary Data 1g).

PCA of the proteomic data revealed differences according to clinical severity (Supplementary Fig. 8a). We found 148 and 1625 proteins associated (LMM, 1% FDR) with disease severity in the Wave 1 (86 COVID-19 positive samples) and Wave 2 (102 COVID-19 positive samples) datasets, respectively (Supplementary Data 1h, Supplementary Fig. 9). 98 proteins were associated with severity in both datasets. Pathway analysis identified 15 severity-associated pathway terms that reached statistical significance (1% FDR) in both cohorts (Supplementary Fig. 8b, Supplementary Data 1l). Among the most upregulated pathways in more severe disease were 'HDACs deacetylate histones', pathways related to transcriptional regulation (e.g., 'mRNA splicing minor pathway', 'Spliceosome', 'RNA polymerase II transcription termination', 'Processing of capped intron-containing pre- mRNA') and 'RUNX1 regulates genes involved in megakaryocyte differentiation and platelet function', while the most down-regulated pathways included 'PD-1 signalling' and 'T-cell receptor and costimulatory signalling'. Example proteins from these pathways are shown in Supplementary Fig. 8c.

## Severe COVID-19 is associated with dynamic multi-omic modular trajectories

We next examined the temporal trajectories of the transcriptome and the proteome during COVID-19 by explicitly modelling molecular profiles with respect to time following symptom onset (Methods). To aid biological interpretation, we first applied a dimension reduction strategy using weighted gene correlation network analysis (WGCNA)[34]. WGCNA identified 23 modules of co-expressed genes (which we denote with the prefix t) (Supplementary Data 1j), and 12 proteomic modules (denoted with p) (Supplementary Data 1k). Longitudinal modelling revealed 8 transcriptomic and 5 proteomic modules with significantly (5% FDR) different temporal patterns in patients with mild/moderate versus severe/critical disease (LMM time x clinical course (TxCC) interaction−Methods) (Supplementary Tables 3-4). Typically, the modules displayed a flat temporal profile in mild/moderate COVID-19, whereas there was a dynamic profile in severe/critical disease (Fig. 4a, Supplementary Fig. 10). Some modules rose with time in severe/critical patients (e.g., tB, tL, p9 and p12), while others dropped (e.g., tC, tP, tI, p7). Examples of individual genes from module tB exhibiting this behaviour include *MMP9*, *ORM1*, *LRRN1* (Fig. 4b).

We identified significant associations between modules, with transcriptomic and proteomic modules clustered into larger positively or negatively correlated groupings (Fig. 4c). The inter-modular associations appeared to strongly reflect association with COVID-19 severity at time of sampling (Supplementary Tables 3-4), implying that this is a strong underlying factor in the -omics data. Consistent with this, integrated analysis of the transcriptomic and proteomic datasets using MEFISTO[35] revealed a single factor that had a significantly different trajectory in severe/critical versus mild/moderate disease ($p = 5.4 \times 10^{-12}$, LMM TxCC) (Supplementary Fig. 11).

We characterised the modules by pathway analysis (Fig. 4a, Supplementary Tables 3-4, Supplementary Data 1l, Supplementary Data 1m). We also investigated whether disease trajectory-associated transcriptomic modules might reflect a shift in cell-type proportions, estimated using the CIBERSORTx algorithm (Methods) (Supplementary Fig. 12, Supplementary Data 1n). The severity-associated modules tB and tJ were both strongly positively associated with myeloid cell proportions, particularly neutrophils, and negatively associated with lymphocyte subsets (Supplementary Fig. 12). The presence of a neutrophilic gene signature in the PBMC preparations may indicate the presence of low-density granulocytes. Consistent with this, hub genes in Module tB (including *TECPR2*, *CSF3R*, *STX3*; Fig. 4a) are associated with granulocytes and autophagy, and pathway analysis of the module genes revealed enrichment for pathways including 'Neutrophil degranulation' and 'ROS and RNS production in phagocytes' (including genes encoding the key cytosolic components of the phagocyte NADPH oxidase such as *NCF1*, *NCF2* and *NCF4*). Module tB also contains genes encoding calcium-binding proteins (e.g., *S100A6*, *S100A9*, *S100A11, S100A12*) that play important roles in regulating inflammatory pathways[36], as well as integrins (e.g., *ITGA1*, *ITGAM*, *ITGB4*, *ITGAX*, *ITGAD*), adhesion molecules (e.g., *CEACAM1*, *CEACAM3*, *CEACAM4*, *ICAM3*), *OSM* (encoding Oncostatin M) and *CSF1* (encoding M-CSF). The tL module, which also displayed a rising trajectory in worse disease, was strongly positively associated with imputed plasma cell proportion (Supplementary Fig. 12) and many of its members encoded immunoglobulins. The severity-associated proteomic modules that strongly correlated with transcriptomic modules tB, tJ and tL were p8 and p9 (both enriched for pathways related to RNA splicing), and p12 (significantly enriched for the pathway 'HDACs deacetylate histones') (Supplementary Table 4). The latter is consistent with our earlier observations that a histone pathway signature was prominently associated with COVID-19 severity in both the RNA-seq (Fig. 3c, Supplementary Fig. 7) and plasma proteomic data (Supplementary Fig. 8c).

In contrast to tB, tJ, and tL, the other transcriptomic modules (tP, tC, tF, tI, tN) all displayed a decreasing trajectory in patients

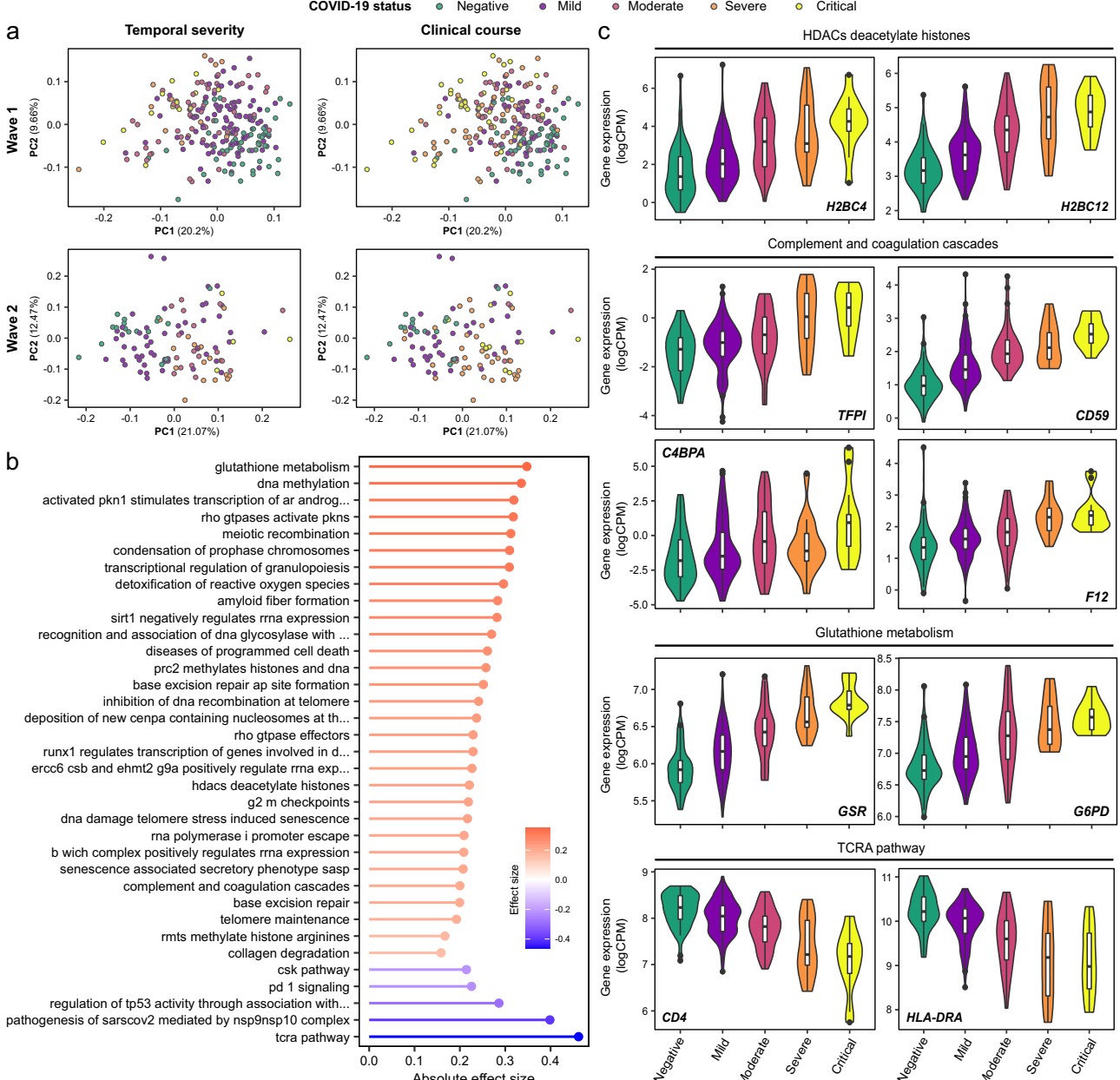

**Fig. 3 | Association of the PBMC transcriptome and COVID-19 severity. a** PCA of the PBMC transcriptome. Each point represents a sample and is coloured by contemporaneous COVID-19 WHO severity (left) and overall clinical course (right). **b** The 30 GSVA transcriptomic pathway enrichment terms with the greatest RRA scores. All were significantly enriched in both Wave 1 and 2 cohorts (1% FDR). Terms are ordered and coloured by their effect size. Red=up-regulated in more severe COVID-19; blue=down-regulated. **c** Violin plots show gene expression values (Wave 1 cohort; n = 179 samples from 51 COVID-19 positive ESKD patients and 55 samples from COVID-19 negative ESKD patients) stratified by COVID-19 status and severity

(at time of sample) for selected genes. Shaded areas indicate kernel density estimates. For boxplots, centre=median, upper bound=upper quartile (UQ), lower bound=lower quartile (LQ), upper whisker=largest value at most 1.5 * IQR (interquartile range) from the UQ, lower whisker=smallest value at most 1.5 * IQR from the LQ, points=samples outside of the range of the whiskers. All genes shown were significantly associated (1% FDR) with severity in both the Wave 1 and 2 cohorts. Genes are grouped by membership to pathways that were significantly enriched (1% FDR) in GSVA.

with worse disease (Fig. 4a). These transcriptomic modules tended to be positively associated with imputed lymphocyte subset proportions and negatively associated with imputed myeloid proportions, implying that higher lymphocyte-related gene signatures and lower myeloid-related ones is a favourable prognostic sign (Supplementary Fig. 12). Our findings are consistent with studies in non-ESKD COVID-19 cohorts that show that a reduction in lymphoid cell proportion and an increase in myeloid cell proportion are associated with more severe disease (e.g.,[8,37]).

**Flow cytometry identifies markers of enhanced interferon signalling early in severe disease**

To understand whether transcriptional signatures in PBMCs reflected changes in blood cell proportions, we performed flow cytometry on a subset of PBMC samples from the Wave 2 cohort. We found no major difference in the overall proportions of myeloid or lymphoid cells within the PBMC fraction between pre-infection and COVID-19 positive samples, except for a reduction in the proportion of type 2 dendritic cells (Supplementary Fig. 13). Similarly, there was little difference in the

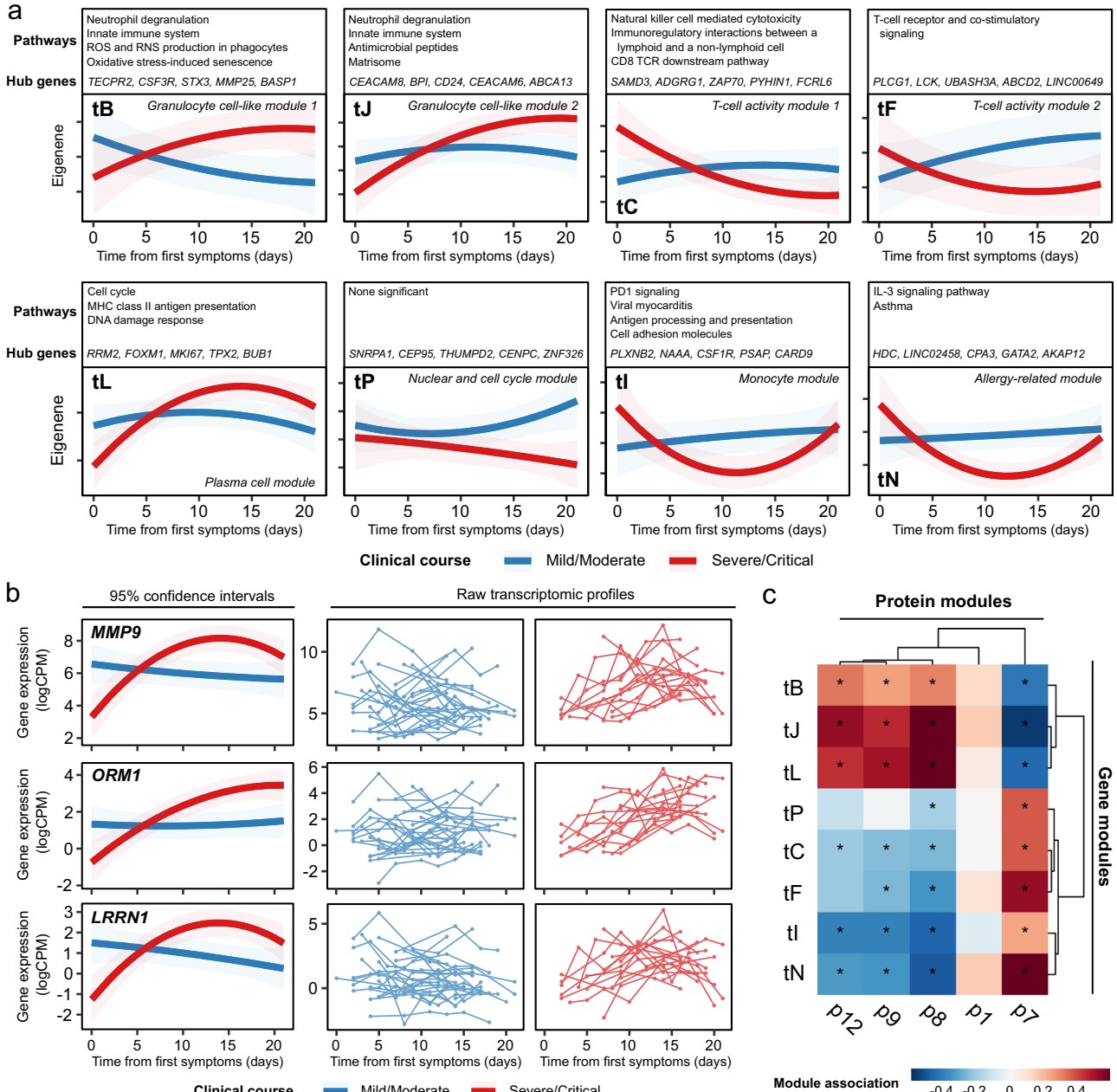

**Fig. 4 | Longitudinal profiles of transcriptomic modules. a** The longitudinal profiles of significant (TxCC, LMM, FDR < 0.05) gene modules, stratified by clinical course. Lines represent estimated marginal means and shaded areas represent their 95% confidence intervals. **b** Modelled longitudinal profiles of the three genes within module B with the most significant TxCC interaction effects (LMM). Left: lines represent estimated marginal means and shaded areas represent their 95% confidence intervals. Right: individual-level data ($n$ = 169 samples from 40 individuals). **c** Heatmap displaying associations (LMM) between transcriptomic and proteomic modules (right). Red=positive correlation, blue=negative correlation. Significant associations (5% FDR) are represented by an asterisk.

distribution of cells between mild/moderate and severe/critical patients. We observed some severity-related differences within cell subsets. Within lymphoid cells, we noted higher expression of the activation marker CD69 on CD4+ T cells at day 7 in severe/critical disease compared to either pre-infection or mild/moderate disease (Supplementary Fig. 14a). At day 14, there was an increase in CD38^hi plasmablasts in severe/critical disease compared to pre-infection or mild/moderate samples (Supplementary Fig. 14b). We also found that in severe/critical patients, there was a progressive drop in the proportion of non-classical monocytes over the first 14 days of the illness that was more marked than in mild/moderate patients (Supplementary Fig. 15a). This is consistent with previous studies in non-ESKD cohorts showing an association between a decrease in non-classical monocytes

and more severe COVID-19 (e.g.,[8,31,38,39].). In severe/critical patients there was a greater proportion of intermediate and non-classical monocyte subsets expressing CD38 compared both to pre-infection samples and to mild/moderate patients (Supplementary Fig. 15b), likely reflecting enhanced activation[30]. In classical monocytes there was a similar, but non-significant, trend. We found higher expression of proliferation-associated Ki67 on classical monocytes in COVID-19 versus pre-infection samples in both mild/moderate and severe/critical patients (Supplementary Fig. 15c). In our transcriptomic data we identified increased *SIGLEC1* gene expression in COVID-19 (Fig. 2b). SIGLEC-1 is exclusively expressed by CD14 + monocytes at the protein level. SIGLEC-1 expression measured by flow cytometry correlated with GSVA enrichment score of type I IFN signatures (Supplementary

Fig. 15d). We observed SIGLEC1 expression increased at greater intensity as early as day 0-3 post infection in severe/critical versus mild/moderate patients, suggesting stronger and a more immediate type I IFN response in severe COVID-19 (Supplementary Fig. 15e).

### Longitudinal cytokine/chemokine analysis reveals distinct temporal profiles that distinguish disease severity

Many plasma proteins associated with severe COVID-19 are canonically intra-cellular proteins. Their elevation in severe COVID-19 may therefore be a readout of increased cell turnover, death, stress, and viral hijacking of host cellular machinery. Consequently, we performed a more focussed analysis examining proteins whose primary biological role is to act extra-cellularly (e.g., cytokines, chemokines, growth factors and their receptors). These classes of proteins are important therapeutic targets in inflammatory diseases[40]. Accordingly, we modelled the temporal profiles of 232 proteins that fell within the KEGG pathway 'Cytokine-cytokine receptor interaction'. Fifty proteins had significantly different profiles in patients with a severe/critical clinical course versus those with mild/moderate ones (TxCC interaction effect, 5% FDR; Supplementary Data 1o). Proteins exhibited distinct patterns of divergence between severe/critical and mild/moderate disease over time (Fig. 5a). Some (e.g., IL1β, IL6, IL15RA, CCL2) showed a relatively stable temporal profile in mild/moderate patients but rising trajectories in severe/critical patients (Fig. 5b). Others (e.g., CCL15, TNFSF13B (BAFF), PDGFRB, EDAR, IFNA10, IFNA13, IFNA16, IFNE, and IFNL3) were elevated early in the disease course and decreased over time, but displayed more marked initial elevations in severe/critical patients (Fig. 5c). Yet other proteins displayed temporal profiles in mild/moderate patients that were inverted compared to severe/critical. For example, CD40LG, TNFSF10 (TRAIL) and IL11 were reduced in the severe/critical versus the mild/moderate group at early timepoints but increased in severe/critical patients later (Fig. 5d). Conversely, leptin, INHBA (inhibin A), and CCL22 were initially higher in severe/critical than mild/moderate patients but with the reverse pattern later on (Fig. 5e). These data illustrate the dynamic nature of the soluble protein response and how this varies according to disease severity, highlighting the limitation of studies that use a single snapshot. Comparison of our data to another study utilising longitudinal proteomic profiling (Filbin et al.[12]) in COVID-19 in a more general patient population revealed similar findings (Supplementary Material), suggesting that the effects we observed are generally not specific to ESKD. One exception was EPOR (erythropoietin receptor). In our ESKD patient data, this exhibited a significantly different temporal profile in severe/critical versus mild/moderate disease ($p = 1.7 \times 10^{-8}$, LMM time x disease course interaction). In contrast, the temporal profile was not significantly different among differing disease severity strata in the data of Filbin et al. ($p = 0.47$, LMM) (Supplementary Material, Supplementary Fig. 16). Erythropoietin (EPO) is a hormone produced by the kidney that promotes red cell formation. In ESKD there is loss of EPO production by the kidney, and consequently patients require exogenous administration of recombinant EPO. The contrasting EPOR longitudinal profiles between severe and mild patients in our cohort are likely to reflect the changes in erythropoietin responsiveness that accompany critical illness and sepsis in ESKD patients.

### Do immune cell transcriptomic and plasma proteomic signatures of COVID-19 differ between ESKD patients and non ESKD patients?

We next sought to investigate whether the plasma proteomic and PBMC transcriptomic signatures that we observed in our data were specific to ESKD patients, or were like those identified in other non-ESKD patient cohorts. To this end, we compared our RNA-seq results to the COvid-19 Multi-omics Blood ATlas (COMBAT) Consortium study[7]. We re-analysed their RNA-seq data so that the analytical approach was as similar as possible to that used in the present study

(see Supplementary Material). For the differential gene expression analysis between COVID-19 positive and negative samples, we found a high degree of concordance between our study and the COMBAT study (Pearson $r$ 0.7 for comparison of estimated effect sizes; Supplementary Fig. 17), despite the use of whole blood in the COMBAT study versus PBMC in our study. We observed similar consistency in the pathway-level GSVA analysis (Supplementary Fig. 17). Our results for the association of gene expression with COVID-19 severity were also generally consistent with the COMBAT study ($r$ 0.6 for gene-level analysis; Supplementary Fig. 18; see Supplementary Material for more detail). These findings suggest that similar immune cell transcriptomic patterns occur in COVID-19, irrespective of whether the patient has ESKD or not.

We then compared our plasma proteomic data to those of Filbin et al.[12], which also used the SomaScan proteomic platform and collected samples at multiple timepoints during acute COVID-19. While there was still some correlation of effect estimates, this was generally lower than with the cross-study transcriptomic comparisons (Supplementary Material; Supplementary Figs. 19 and 20). This may indicate a greater impact of ESKD on the circulating proteome in COVID-19 than the immune cell transcriptome, although we cannot exclude the possibility that technical factors account for the differences.

In summary, these comparisons reveal that our results are generally similar to those identified in cohorts of COVID-19 patients without ESKD, albeit less so for plasma proteomics than for RNA-seq. Despite the lack of clear systematic differences in the -omic signatures in ESKD patients and other cohorts, manual review of our significant results did reveal specific instances of some biologically plausible examples of ESKD-specific effects, such as the example of EPOR described above.

### Plasma LRRC15 as a marker of COVID-19 severity

We next investigated whether clinical severity could be inferred from the transcriptomic and/or proteomic data and which had the better predictive performance. For each COVID-19 patient, we selected the first sample at the patient's peak severity score so that there was one sample per patient. To predict COVID-19 severity at time of sampling, we employed two supervised learning methods, lasso and random forests. We applied these separately on (i) the plasma proteomic data; (ii) the PBMC transcriptomic data; and (iii) the combination of both (the multi-omic data). For this analysis, we combined the COVID-19 cases from both cohorts. Area under the curve (AUC) was estimated using Monte Carlo cross-validation (Methods). As an additional analysis, we trained models on the Wave 1 cohort and tested on the Wave 2 cohort (Supplementary Material).

The proteomic-based models outperformed the transcriptome-based ones, with non-overlapping 95% confidence intervals (Fig. 6a, Supplementary Fig. 21a). The lasso model generated on the proteome had an estimated AUC of 0.93 (versus 0.86 for the transcriptome). The random forests model generated on the proteome had an AUC of 0.88 (versus 0.83 for the transcriptome). The models based on the proteome alone also had greater predictive performance than those trained on the multi-omic data, although the confidence intervals for the AUC estimates overlapped (Fig. 6a, Supplementary Fig. 21a).

We next examined the supervised learning models to identify the most important biomarkers of severe/critical disease (Methods) (Fig. 6b, Supplementary Fig. 21b, Supplementary Data 1p-r). Although only a minority of the input features to the multi-omic model were proteins (34%; 6323/18,548), proteins made up the majority of the top 15 most important predictors (10/15 for lasso and 9/15 for random forests). This, and our finding that the plasma proteome was a superior classifier of severity than the PBMC transcriptome, highlights that plasma proteins provide a valuable read-out of the pathophysiological processes in severe COVID-19.

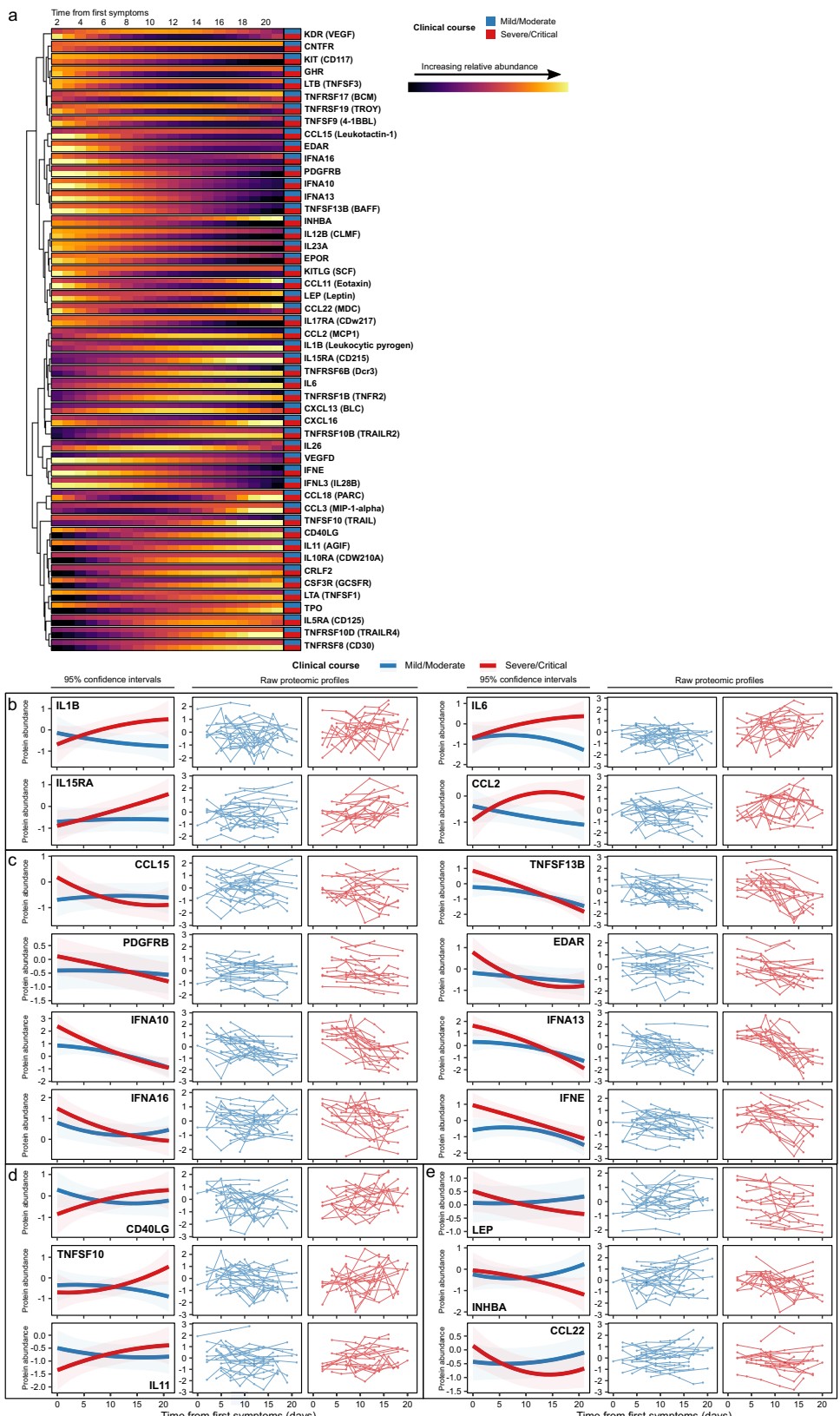

**Fig. 5 | Dynamic temporal changes in circulating cytokines and receptors vary between severe and mild COVID-19. a** Heatmap displaying proteins with a significantly different temporal profile in mild vs severe disease (TxCC, LMM, FDR < 0.05). Colour indicates LMM estimated marginal means over time, stratified by patient group (*n* = 169 samples from 40 individuals). Proteins are clustered based on the temporal profile of the discordance between mild/moderate and severe/critical disease. Proteins are annotated using gene symbols, with alternative common protein identifiers in parentheses. **b**–**e** Examples of proteins with differing patterns of discordance over time in severe/critical versus mild/moderate patients (TxCC, LMM, FDR < 0.05). Lines represent estimated marginal means and shaded areas represent their 95% confidence intervals. The raw proteomic profiles display protein abundance for each individual (*n* = 169 samples from 40 individuals).

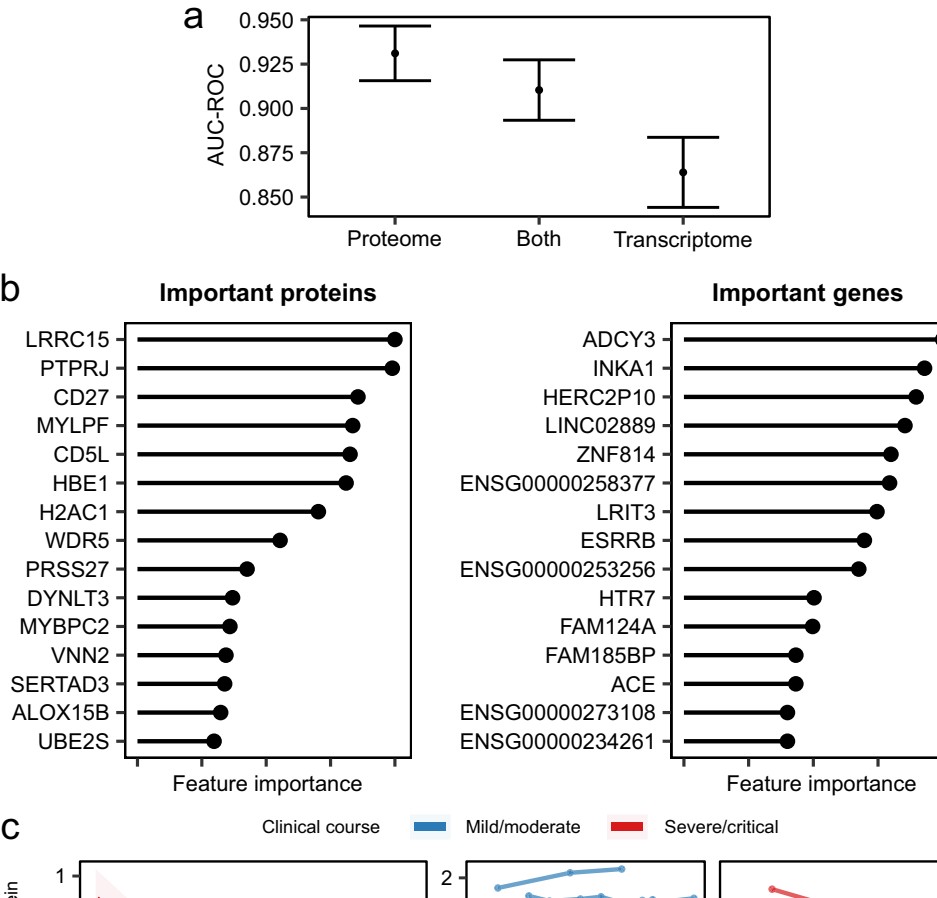

**Fig. 6 | Supervised learning to predict COVID-19 severity from molecular features. a** Point estimates (mean) and 95% confidence intervals of the area under the receiver operating characteristic curve (AUC-ROC) for predicting COVID-19 severity (from 200 cross-validation resamples of 51 independent samples) using lasso regression. Both = supervised learning on the combined proteomic and transcriptomic data. **b** Important proteins (left) and genes (right) for the lasso model. Feature importance is scaled between 0 and 1, where 1 represents the most important feature. **c** The profile of LRRC15 plasma protein concentration over time, stratified by severity of the patients' overall clinical course ($n = 169$ samples from 40 individuals). Left: lines represent estimated LMM marginal means and shaded areas represent their 95% confidence intervals. Right: raw data for each individual.

Both lasso and random forests identified plasma LRRC15 protein levels as the most important biomarker of COVID-19 severity. Interestingly, this protein was recently identified by three pre-prints as a receptor for SARS-COV-2[41–43]. We next examined LRRC15's longitudinal trajectory over the course of COVID-19 infection, finding that it displayed a different temporal profile dependent on the disease course ($p = 6.5 \times 10^{-8}$, TxCC interaction, LMM). The concentration was stable in most individuals with mild/moderate COVID-19 (Fig. 6c), whereas it decreased over time in severe/critical patients. Thus, a snapshot level of LRRC15 and its dynamic profile over time can convey information on the current clinical state of the patient and the overall course of the disease, respectively. Of note, data mining of external studies revealed similar findings in two other studies[12,19] that measured LRRC15 in non-ESKD COVID-19 patients (Supplementary Material, Supplementary Fig. 22).

**Persistent deranged platelet and coagulation pathways in convalescence**

For 12 of the 17 patients in the Wave 2 cohort, we obtained a sample after clinical recovery at approximately two months following the

acute infection. PCA analysis of the PBMC transcriptome showed that while pre-COVID-19 and convalescent samples appeared more similar than samples taken during COVID-19, there were differences between the convalescent samples and their pre-infection counterparts (Fig. 7a), indicating that they have not fully returned to baseline. Comparison of the convalescent samples to their paired pre-COVID-19 samples revealed 25 significantly differentially expressed genes (1% FDR), of which 24 were up-regulated post-COVID-19 (Fig. 7b, Table 1, Supplementary Data 1s). Up-regulated clotting-related genes included *PF4* (encoding platelet factor 4) and the related gene *PF4V1* (platelet factor 4 variant 1). Of note, these genes are located in the same genomic region on chromosome 4, along with the chemokine *CXCL5*, which was also significantly up-regulated. Another nearby gene, *PPBP* (encoding Pro-Platelet Basic Protein, also known as CXCL7), was also up-regulated in convalescent samples, although it did not quite reach significance at 1% FDR (nominal $P = 3.3 \times 10^{-5}$, Benjamini-Hochberg adjusted $P = 0.016$). The upregulation of these neighbouring genes suggests they are influenced by a shared genomic regulatory element. Overrepresentation analysis of the 25 differentially expressed

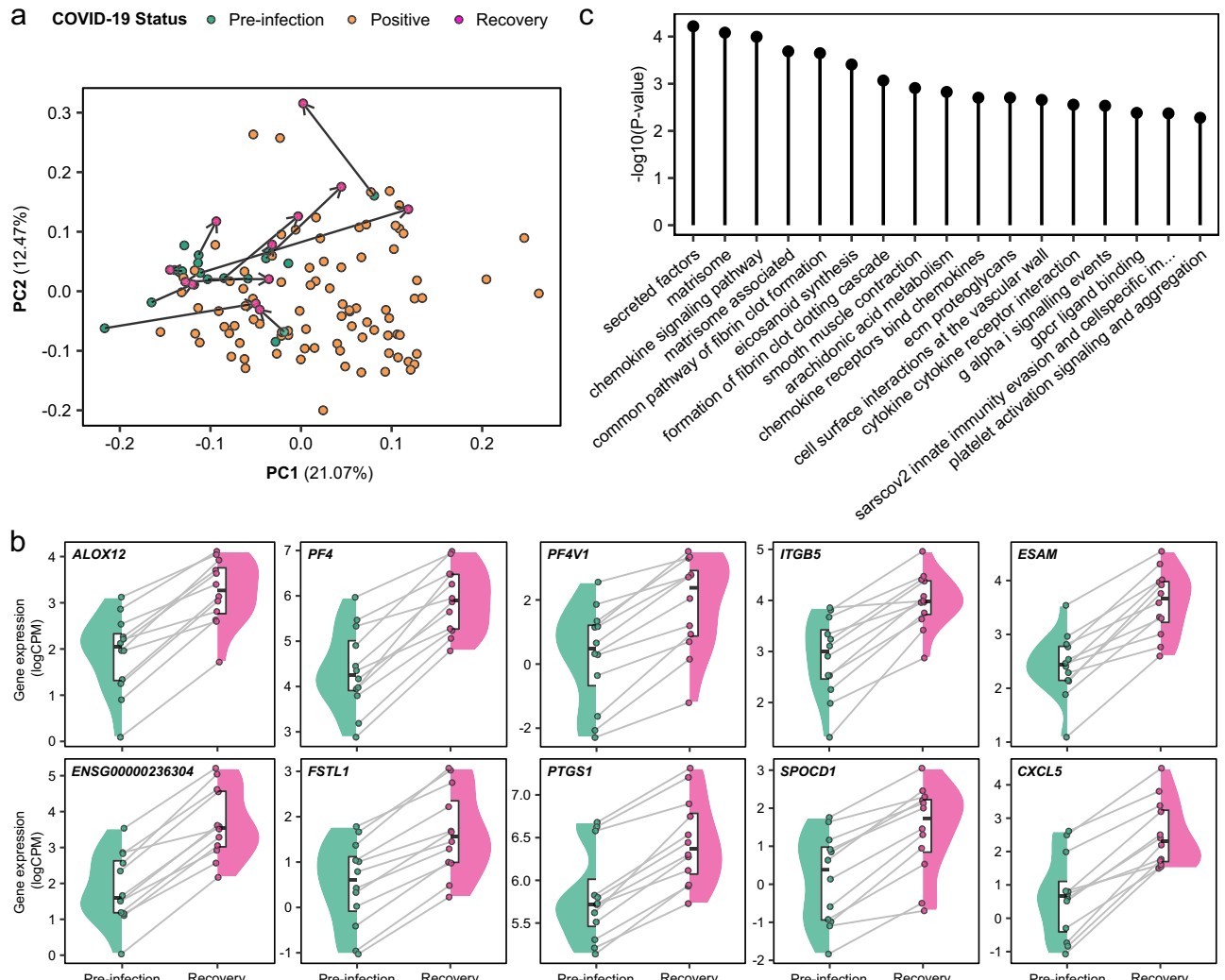

**Fig. 7 | Persistent dysregulation of immune cell gene expression two months following COVID-19. a** PCA of the Wave 2 PBMC transcriptomic data, including pre-infection, infection and recovery samples (taken 2 months after the acute illness). Each point represents a sample. Arrows link recovery samples to the pre-infection sample from the same individual. **b** Paired violin plots for differentially expressed genes in recovery versus pre-infection samples (*n* = 24 samples from 12 individuals). Grey lines link each individual's pre-infection sample to their recovery sample; these samples are represented by points. Shaded areas indicate kernel density estimates. For boxplots, centre=median, upper bound=upper quartile, lower bound=lower quartile. **c** All significantly enriched (5% FDR) pathway terms for the differentially expressed genes in recovery versus pre-infection samples. Over-representation testing was performed for each gene set with a one-sided Fisher's exact test.

genes revealed significant enrichment of terms including 'Platelet activation, signalling and aggregation', 'Formation of fibrin clot/clotting cascade', 'Chemokine signalling pathway', 'SARS-CoV-2 innate immunity evasion and cell-specific immune response' and 'Smooth muscle contraction' (Fig. 7c, Supplementary Data 1t). These data suggest persistent activation of abnormal processes for a considerable time after clinical recovery. In particular, they implicate the vascular and clotting systems, which may have implications for long-term risk of thrombosis.

## Discussion

Here we performed serial blood sampling and longitudinal multi-omic analysis of ESKD haemodialysis patients with COVID-19, enabling insight into the pathogenesis of COVID-19 through examination of the temporal evolution of molecular and cellular changes. ESKD patients are an important group to study as they are at elevated risk of severe or fatal disease[25,44]. Despite the remarkable success of vaccination programmes at the population level, ESKD patients display impaired vaccine responses[27,28]. In addition, the majority of patients in our study

were of non-white ethnicity, which is also a risk factor for severe disease[25].

Most studies of circulating proteins in COVID-19, including our previous work, have used Olink immunoassay technology[12–14] or mass spectrometry[15,16]. The broadest Olink assay system, used in the study of Filbin et al.[12], measures 1472 proteins, while mass spectrometry is generally limited to reliable detection of fewer than 1000 plasma proteins and lacks sensitivity for low abundance proteins. A small number of studies have employed the aptamer-based SomaScan v4 platform, that measures 4665 unique proteins[12,17–20]. Here, we used the SomaScan v4.1, which measures 6323 unique proteins, and complemented this with RNA-seq and flow cytometry. Our study is strengthened by data from two cohorts from different waves of the pandemic, and the comparison of samples from before, during and after COVID-19 from the same individuals.

Plasma proteomics identified several pathways upregulated in COVID-19 related to host defence against viruses, including those previously described in SARS-CoV-2. Our PBMC transcriptomic analysis identified numerous pathways that are up-regulated in COVID-19.

**Table 1 | Genes that do not return to baseline 2 months after recovery from COVID-19**

| Gene ID | Gene name | Estimate (recovery - pre-infection) | P-value | BH-adjusted P-value |
|---|---|---|---|---|
| ENSG00000236304 | lncRNA | 1.84 | 5.37E-08 | 9.20E-04 |
| FSTL1 | Follistatin Like 1 | 1.17 | 1.60E-07 | 1.37E-03 |
| PTGS1 | Prostaglandin-Endoperoxide Synthase 1 | 0.64 | 3.69E-07 | 2.11E-03 |
| SPOCD1 | SPOC Domain Containing 1 | 1.31 | 6.41E-07 | 2.53E-03 |
| CXCL5 | C-X-C Motif Chemokine Ligand 5 | 1.84 | 7.39E-07 | 2.53E-03 |
| ALOX12 | Arachidonate 12-Lipoxygenase, 12 S Type | 1.33 | 1.24E-06 | 3.53E-03 |
| PF4 | Platelet Factor 4 | 1.53 | 1.71E-06 | 3.91E-03 |
| ESAM | Endothelial Cell Adhesion Molecule | 1.18 | 1.83E-06 | 3.91E-03 |
| MT-RNR1 | Mitochondrially Encoded 12 S RRNA | 1.00 | 2.61E-06 | 4.13E-03 |
| MMD | Monocyte To Macrophage Differentiation Associated | 0.77 | 2.80E-06 | 4.13E-03 |
| ENSG00000240093 | lncRNA | -0.71 | 2.82E-06 | 4.13E-03 |
| MTURN | Maturin, Neural Progenitor Differentiation Regulator Homolog | 0.54 | 3.11E-06 | 4.13E-03 |
| GNG11 | G Protein Subunit Gamma 11 | 1.53 | 3.13E-06 | 4.13E-03 |
| CAVIN2 | Caveolae Associated Protein 2 | 1.15 | 4.51E-06 | 5.52E-03 |
| DOK6 | Docking Protein 6 | 1.59 | 5.00E-06 | 5.71E-03 |
| LINC00989 | lncRNA | 1.25 | 5.76E-06 | 5.81E-03 |
| SPARC | Secreted Protein Acidic And Cysteine Rich | 1.65 | 6.07E-06 | 5.81E-03 |
| PF4V1 | Platelet Factor 4 Variant 1 | 1.60 | 6.11E-06 | 5.81E-03 |
| ABLIM3 | Actin Binding LIM Protein Family Member 3 | 1.41 | 6.88E-06 | 6.07E-03 |
| MFAP3L | Microfibril Associated Protein 3 Like | 0.74 | 7.09E-06 | 6.07E-03 |
| CALD1 | Caldesmon 1 | 1.78 | 7.79E-06 | 6.23E-03 |
| ITGB5 | Integrin Subunit Beta 5 | 1.10 | 8.00E-06 | 6.23E-03 |
| LINC01750 | lncRNA | 2.00 | 1.13E-05 | 8.22E-03 |
| PCSK6 | Proprotein Convertase Subtilisin/Kexin Type 6 | 1.22 | 1.15E-05 | 8.22E-03 |
| PVALB | Parvalbumin | 2.04 | 1.46E-05 | 9.97E-03 |

Genes that are significantly differentially expressed (1% FDR, LMM) in recovery versus pre-infection samples ($n = 24$ samples from 12 patients). Estimate represents the estimated difference in log CPM (counts per million). BH=Benjamini-Hochberg. A complete table including all genes tested can be found in Supplementary Data 1s.

Many have been identified in previous studies of COVID-19 in other populations without ESKD, indicating the presence of common patterns of COVID-19-related immunological abnormalities. Examples include type 1 interferon signalling, the complement cascade, and genes reflecting leukocyte-vascular interactions. Other up-regulated pathways included 'Polo-like kinase mediated events' and 'Golgi-cisternae peri-centriolar stack re-organisation'. Both are likely to reflect the extensive cell division of immunocytes that occurs in COVID-19. For instance, the pericentriolar stacks of Golgi cisternae undergo extensive fragmentation and reorganization in mitosis. Similarly, polo-like kinase is crucial for facilitating the G2/M transition. These findings are consistent with the up-regulation of APC-Cdc20 mediated degradation of Nek2A and other APC-Cdc20 related processes that we observed in the proteomic data; Cdc20 is a protein that is key to the process of cell division.

Transcriptomic and proteomic associations with severe COVID-19 converged on some unifying themes, including enrichment of pathways related to histones, interferon response, granulopoiesis, clotting, TCR activation and cell cycle processes. For example, up-regulation of histone-encoding genes and elevated plasma histone protein levels were both markers of COVID-19 severity. The increased expression of histone-encoding transcripts may indicate increased immune cell proliferation. In each cell cycle, sufficient histones are needed to package the newly replicated daughter DNA strands, requiring tight coupling of histone synthesis to the cell cycle[45]. Excess histones within cells can trigger chromatin aggregation and block transcription[46]. Thus, in severe COVID-19, viral hijacking of cellular machinery may contribute to cellular damage through decoupling of DNA synthesis and histone transcription. The preponderance of plasma histone proteins in severe disease is likely to reflect the higher levels of cell

damage and death. The presence of histone proteins in plasma, however, may represent more than just a marker of disease. Histones are constituents of neutrophil extracellular traps (NETs) which contribute to tissue injury in severe COVID-19. In addition, histones constitute powerful damage associated molecular patterns (DAMPs) and can perpetuate inflammation via ligation of toll-like receptors and direct damage to epithelial and endothelial cells[47]. Upregulation of pathways related to control of transcription and translation was another feature of severe COVID-19 (Supplementary Fig. 8b), perhaps reflecting subversion of normal cell biology by SARS-CoV-2. In keeping with this, studies of cells infected with SARS-CoV-2 revealed alteration of processes including translation, splicing and nucleic acid metabolism[48,49].

Modular analysis highlighted a rising neutrophilic gene signature as the illness progressed in severe/critical patients, with enrichment of reactive oxygen and nitrogen species pathways. This suggests prolonged activation of neutrophils and their key effector pathways including NET formation. This neutrophilic gene signature likely indicates the presence of low-density granulocytes within the PBMC fraction. Data from other infections suggest that phagocyte NADPH oxidase-derived reactive oxygen species can be detrimental in acute viral infection; mice lacking components of the NADPH oxidase have reduced disease severity and inflammation in response to influenza and lymphocytic choriomeningitis virus infection[50–52].

Cytokines and their receptors play a major role in the pathogenesis of inflammatory diseases and are important targets of existing drugs[40]. Longitudinal examination of plasma cytokines/chemokines revealed divergence temporal trajectories between disease severity strata, manifesting in several patterns (Fig. 5). For example, in patients with a severe/critical disease course, IL11 was reduced early on but increased later relative to more indolent disease (Fig. 5d). IL11 is known

to cause progressive fibrosis[53,54], and the marked increases late in severe/critical disease may have implications for the development of pulmonary sequelae. Leptin, INHBA (inhibin A), and CCL22 showed the opposite pattern (Fig. 5e). Leptin has roles in both cell metabolism and immunity with many immune cells responding to leptin directly via the leptin receptor, resulting in a pro-inflammatory phenotype[55]. It is produced by adipocytes, so its elevation early in severe/critical disease may be a read-out of higher body mass index, which is a risk factor for severe COVID-19, or increased cell metabolism/turnover. Its fall over time in severe/critical patients may reflect weight loss and cell death. Whether leptin is also directly influencing risk of severe disease through its immunological effects is unclear. Inhibin-A progressively increased over time in mild/moderate patients but fell in severe/critical patients. Inhibin-A negatively regulates dendritic cell maturation and promotes a tolerogenic phenotype[56]. Failure to upregulate it later in the disease course may therefore contribute to deleterious inflammation. Similarly, CCL22 plays an important role in switching off inflammation. CCL22 promotes dendritic cell-regulatory T cell interactions and CCL22 deficiency is associated with excessive pathogenic inflammation in mice[57].

Proteins in the type 1 interferon (IFN) pathway were higher in severe/critical than mild/moderate patients early in disease (Fig. 5c), suggesting a paradoxical role of this pathway in COVID-19. While inherited or acquired deficiencies of IFN proteins predispose to risk of severe COVID-19[58,59], our data suggest that the picture may be more complex. Thus, IFNs may act as a double-edged sword, with harm to the host from both insufficient responses (leading to failure to control the virus) and from excessive responses (resulting in immunopathology). While we cannot exclude the possibility that increased IFNs is a consequence rather than a cause of severe disease, their elevation very early in disease suggests this is less likely. Another consideration is that the greater IFN response in severe disease might reflect higher viral burden.

Using two distinct supervised learning methods, we observed that the plasma proteome better captures disease severity than the PBMC transcriptome. When supervised learning algorithms were trained on both the proteomic and transcriptomic data simultaneously, plasma proteins dominate the list of important biomarkers. There are several reasons why this might be the case. Plasma is under strong homeostasis: derangement is a marker of loss of physiological control. Plasma proteins may provide important read-outs of both pathogenesis and tissue injury by reflecting the activity of cell types other than PBMCs, such as neutrophils, endothelium and hepatocytes (a major source of coagulation and complement proteins). In apparent contrast to our findings, a study by Lee et al.[9] involving immune cell transcriptomics and plasma metabolomics using mass spectrometry (MS) reported that the combination of transcriptome and metabolome provided superior classification of severity. However, it is likely that the difference between the study of Lee et al. and our findings relate to what was measured (i.e., MS-based metabolomics versus measurement of 6,323 proteins).

Our integrated multi-omics analysis with MEFISTO revealed a single factor that had a significantly different trajectory in severe/critical versus mild/moderate disease (Supplementary Fig. 11a). This parallels the findings of Su et al.[8]. While the specific methods used differ, both our MEFISTO analysis and the integrative network analysis by Su et al. identified a single factor in the data that was highly related to COVID-19 severity and pro-inflammatory cytokines.

Comparison to other transcriptomic and plasma proteomic studies in non-ESKD patients revealed broadly similar findings. Such inter-study comparisons have inherent limitations as it is not possible to distinguish whether study-specific findings are biological or are due to differences in study design, statistical power, assay platforms and other sources of non-biological variation. With these caveats in mind, our transcriptomic findings were remarkably similar to those of the

COMBAT study[7] (despite PBMC being measured in our study versus whole blood in their study). There was lower concordance of our plasma proteomic results to those of Filbin et al.[12] than for the transcriptomic comparison. This could reflect differences in study design (unlike our controls, those of Filbin et al. presented with acute respiratory distress) or technical differences (the study by Filbin et al. used an earlier version of the SomaScan platform) but could also be biological as it is known that circulating proteins are affected by renal impairment[60–63].

One finding that may be specific to ESKD is the dynamic temporal profile of the erythropoietin receptor (EPOR) in severe/critical COVID-19 versus a more stable profile in mild/moderate disease. This is likely to reflect the changes in erythropoietin responsiveness that accompany critical illness and sepsis in ESKD patients. Cytokines affect the EPO-mediated signalling pathway[64] and inhibit the expression and regulation of specific transcription factors involved in the control of erythrocyte differentiation. For instance, high concentrations of TNF-α or IFN-gamma cause the need for higher amounts of EPO to restore the formation of erythrocyte colony forming units[65]. In haemodialysis patients, inflammation decreases the response to erythropoiesis stimulating agents, changing iron regulation through hepcidin upregulation and facilitating haemolysis[66]. This EPO-hyporesponsiveness (assessed in terms of haemoglobin) in the setting of acute inflammation has been empirically demonstrated in a large multi-national study[67]. Furthermore, patients with ESKD rely on regular exogenous administration of erythropoiesis-stimulating agents and this can be disrupted when patients are admitted to hospital. Dysregulation of the erythropoietin pathway in severe COVID-19 in ESKD patients may also be relevant to immune function since erythropoietin is known to effect both innate and adaptive immunity[68].

A notable finding was the identification of plasma levels of LRRC15 as a marker of COVID-19 severity (Fig. 6b). Longitudinal profiling revealed that LRRC15 levels remain stable in those with a mild/moderate clinical course but decrease over time in severe/critical illness (Fig. 6c). Data mining of previous studies revealed similar findings in two non-ESKD cohorts[12,19]. Three recent pre-prints using a variety of cell lines and approaches have identified LRRC15 as a SARS-CoV-2 co-receptor[41–43]. Using arrayed transmembrane protein and pooled genome-wide CRISPR activation screens, Shilts and colleagues demonstrated that the SARS-CoV-2 spike protein interacts with LRRC15[41]. Both screens identified the interaction and the CRISPRa screen identified LRRC15 and the established SARS-CoV-2 binding partner, ACE2, as the two most prominent interactors. This work also showed that ACE2 and LRRC15 bind the C-terminal domain of the spike protein, which contains the receptor binding domain, suggesting that the two proteins may compete for spike protein binding. Loo et al.[43] performed a CRISPRa screen on HEK293T cells, also identifying LRRC15 and ACE2 as the highest confidence SARS-Cov2 receptors. They propose that LRRC15 plays an actively inhibitory role, binding SARS-CoV-2 but not allowing entry to the cell. They further hypothesise that it does so in trans through its high expression on fibroblasts rather than alveolar cells. Song and colleagues[42] also used a CRISPRa approach to identify proteins that could bind the SARS-CoV-2 spike protein to the A375 melanoma cell line. The screen identified ACE2 and LRRC15, and further showed that the interaction took place with the receptor binding domain of the spike protein. Expression of LRRC15 on a HeLa cell line that expresses ACE2 inhibited the entry of a SARS-CoV-2 spike pseudovirus. This paper also notes that LRRC15 is expressed on different cells from those that express ACE2 and proposes that LRRC15 inhibits virally entry in trans, acting as a decoy and binding virions that cannot then enter cells via ACE2. Our data provide in vivo human evidence to suggest LRRC15 may be important in the host response to SARS-CoV-2, and are consistent with a model in which a failure to upregulate LRRC15 increases risk of severe COVID-19 disease because of the lack of a receptor that inhibits its entry to cells.

A strength of our study was the availability of baseline pre-infection samples for the Wave 2 cohort, as well as samples taken two months after the acute COVID-19 episode. Leveraging this, we demonstrate that there is chronic activation of gene expression related to vascular, platelet and coagulation pathways for a prolonged period after clinical resolution of disease. The elevated risk of thrombotic events during acute COVID-19 is well-documented. In a large study encompassing both hospitalised and non-hospitalised patients[69], the risk of pulmonary embolism (PE) and deep vein thrombosis (DVT) were 27-fold and 17-fold increased, respectively, in the seven days following diagnosis. These risk ratios are much higher than those previously associated with upper respiratory tract infections, suggesting unique features specific to SARS-CoV-2 infection. The risk of arterial thrombosis was also significantly increased, although smaller in magnitude than the risk of venous thromboembolism (VTE). The pathophysiology underlying COVID-19 associated coagulopathy is complex and involves the convergence of several pathways[70]. Invasion of ACE2-expressing epithelial cells by SARS-CoV-2 results in down-regulation of ACE2 and increased angiotensin II levels. This in turn leads to increased expression of PAI1 which impairs breakdown of fibrin and promotes increased vascular tone, via smooth muscle contraction. Endothelial cell activation, complement activation, NETosis, hypoxia and cytokine/chemokine secretion all promote coagulopathy through increases in tissue factor and concomitant fibrin formation. Our data suggest that these pathways remain dysregulated months after acute infection has resolved (Fig. 7, Table 1). This is important given emerging evidence indicating that the risk of thrombo-embolism extends beyond the acute phase. Ho et al. showed that risk of a PE was 3.5-fold higher even in the time window 28 to 56 days after diagnosis of COVID-19[69]. A recent population-wide registry study revealed that following COVID-19 the risk of DVT and PE was significantly elevated for 70 and 110 days, respectively[71]. Although VTE risk was greatest for those with severe disease, even patients with mild disease had elevated VTE risk. Our data provide a molecular basis that begins to explain this risk. Among the genes up-regulated in convalescent samples compared to pre-infection was platelet factor 4 (PF4). PF4 is expressed in platelets and leucocytes. It is released from the alpha granules of activated platelets, contributing to platelet aggregation. The prolonged up-regulation of PF4 after COVID-19 is therefore likely to contribute to a prothrombotic state. Of note, autoantibodies to PF4 are the pathogenic entity in both vaccine-induced thrombotic thrombocytopenia (VITT)[72,73] and heparin-induced thrombocytopenia (HIT). PF4 becomes an autoantigen when it forms complexes with adenoviral vaccine components or heparin respectively, unmasking epitopes to which autoantibodies bind[74]. It will therefore be interesting for future studies to investigate whether autoantibodies to PF4 might contribute to post-COVID-19 thrombosis in some patients. Whether the molecular abnormalities found in our study also apply to more general patient populations without background ESKD needs to be determined. Ongoing studies focusing on the sequelae of COVID-19 are well placed to address this.

Our study has several limitations. It was a single centre study and so lacked a truly independent external validation cohort. ESKD patients have considerable multi-morbidity and deranged physiology, and our findings may not all be generalisable to other patient populations. We lacked a comparator group of ESKD patients with another viral infection to delineate COVID-19 specific features. We studied peripheral blood; while this can provide valuable information, it does not always reflect processes at the site of tissue injury. We performed bulk RNA-seq on PBMCs. Thus, transcriptomic signatures may reflect both changes in gene expression and also alteration in the distribution of cell types within PBMCs. We mitigated this issue through use of deconvolution methods and flow cytometry, but future studies using single cell RNA-seq and CITE-seq will provide further granularity. We did not have measurements of viral load which would have aided interpretation of the magnitude of host responses (e.g., interferon

signalling). Finally, the convalescent samples were taken relatively soon after clinical recovery: it will be important for future studies to establish how long molecular abnormalities persist.

In summary, we demonstrate dynamic transcriptomic, proteomic and cellular signatures that vary both with time and COVID-19 severity. We show that in patients with a severe clinical course there is increased type 1 interferon signalling early in the illness, with increases in pro-inflammatory cytokines later in disease. We identify plasma levels of the proposed alternative SARS-CoV-2 receptor, LRRC15, as a marker of COVID-19 severity. Finally, we show that immune cells display dysregulated gene expression two months following COVID-19, with upregulation of clotting-related genes. This may contribute to the prolonged thrombotic risk post-COVID-19.

## Methods
### Patient cohorts and ethical approval
All participants were recruited from the Imperial College Renal and Transplant Centre and its satellite dialysis units, London, United Kingdom, and provided written informed consent prior to participation. Study ethics were reviewed by the UK National Health Service (NHS) Health Research Authority (HRA) and Health and Care Research Wales (HCRW) Research Ethics Committee (reference 20/WA/0123: The impact of COVID-19 on patients with renal disease and immunosuppressed patients). Ethical approval was given. Study volunteers provided informed consent and did not receive financial or other compensation for participating in the study.

We recruited two cohorts of ESKD patients with COVID-19 (Fig. 1a). All patients were receiving haemodialysis prior to acquiring COVID-19. The first cohort (Wave 1) were recruited during the initial phase of the COVID-19 pandemic (April–May 2020). Blood samples were taken from 53 patients with COVID-19 (Supplementary Table 1). Serial blood sampling was carried out where feasible (Fig. 1b), given the pressure on hospital services and the effects of national lockdown. We also contemporaneously recruited 59 non-infected haemodialysis patients to provide a control group, selected to mirror the age, sex and ethnicity distribution of the COVID-19 cases (Supplementary Fig. 1a–c).

The Wave 2 cohort consisted of 17 ESKD patients with COVID-19 infected during the resurgence of cases in January–March 2021 (Supplementary Table 2). These 17 individuals had all been recruited as part of the COVID-19 negative control group during Wave 1, and so a pre-infection sample collected in April/May 2020 (8–9 months preceding infection) was also available. For the Wave 2 cohort, we systematically acquired serial samples for all patients at regular intervals (every 2–3 days over the course of the acute illness) (Fig. 1c). Additionally, for 12 of these 17 patients, we acquired convalescent samples at approximately 2 months post the acute COVID-19 episode (range 41-55 days from the initial sample). Convalescent samples were unavailable for four patients who died and for one patient due to logistical difficulties in sample collection.

To minimise variation related to the timing of dialysis, blood samples were taken prior to commencing a haemodialysis session.

### Clinical severity scoring
We assessed disease severity using a four-level ordinal score, categorising into mild, moderate, severe, and critical, based on the WHO clinical management of COVID-19: Interim guidance 27 May 2020. Mild was defined as COVID-19 symptoms but no evidence of pneumonia and no hypoxia. Moderate was defined as symptoms of pneumonia or hypoxia with oxygen saturation ($SaO_2$) greater than 92% on air, or an oxygen requirement no greater than 4 L/min. Severe was defined as $SaO_2$ less than 92% on air, or respiratory rate more than 30 per minute, or oxygen requirement more than 4 L/min. Critical was defined as organ dysfunction or shock or need for high dependency or intensive care support (i.e., the need for non-invasive ventilation or intubation). We recorded disease severity scores throughout the illness, such that

samples from the same individual could have differing severity scores according to the temporal evolution of the disease. We defined the overall clinical course for each patient as the peak severity score that occurred during the patient's illness. Different downstream analyses utilise either the severity at the time of sample (i.e., the sample-level severity) or the overall clinical course (i.e., the patient-level severity), as described in the relevant sections below.

### PBMC collection protocol

Peripheral blood mononuclear cells (PBMCs) were obtained by density gradient centrifugation using Lymphoprep (STEMCELL Technologies, Canada). Approximately 20 ml of blood were diluted 1× with phosphate buffered saline (PBS) with addition of 2% FBS and layered on top of 15 ml of Lymphoprep solution. The samples were then centrifuged at 800 g for 20 min at room temperature without break. PBMCs were collected from the interface and washed twice with PBS/2%FBS. In total 2 million PBMCs were centrifuged down to form a pellet and resuspended in 350 μl RLT buffer + 1% β-Mercaptoethanol (from Qiagen RNAeasy kit) for RNA extraction. Remaining PBMCs were cryopreserved in 1 ml freezing medium (FBS 10% DMSO) and stored in −80 °C freezer.

### Plasma collection

5 ml of blood was collected in EDTA tubes and centrifuged at $1000 \times g$ for 15 min. Plasma was extracted and frozen at −80 °C.

### RNA-seq of PBMCs

RNA extraction and sequencing were done at GENEWIZ facilities (Leipzig, Germany). Total RNA was extracted from using RNeasy Mini kits (Qiagen) as per the manufacturer's instructions, with an additional purification step by on-column DNase treatment using the RNase-free DNase Kit (Qiagen) to remove any genomic DNA. Total RNA quality and concentration was analysed using Agilent Tapestation (Agilent Tech Inc.). Samples with RIN values ≥6.0 and ≥100 ng of total RNA were used to generate RNA-seq libraries. RNA-seq libraries were made using NEBnext ultra II RNA directional kit per the manufacturer's instruction. Poly-A RNA was purified using poly-T oligo-attached magnetic beads followed by haemoglobin mRNA depletion using QIAseq FastSelect Globin Kit to remove potential contaminating RNA from red blood cell. Then, first and second cDNA strand synthesis was performed. Next, cDNA 3′ ends were adenylated and adapters ligated followed by library amplification. The libraries were size selected using AMPure XP Beads (Beckman Coulter), purified and their quality was checked using a short sequencing run on MiSeq Nano. Samples were randomized to avoid confounding of batch effects with clinical status and multiplexed libraries were run on 29 lanes of the Illumina HiSeq platform to generate approximately 30 million x 150 bp paired-end reads per sample.

Initial quality control and alignment was performed using the nf-core RNA-seq v3.2 pipeline[75] based on nextflow[76], a workflow management system. FastQC[77] was used to evaluate and merge paired reads prior to adapter trimming using Trimgalore[78]. We used STAR[79] to align reads to GRCh38 and htseq-count[80] to generate a counts matrix.

For the Wave 1 cohort, transcriptomic data were available for 179 samples from 51 COVID-19 positive ESKD patients (median 3 samples per patient, range 1–8) (Supplementary Fig. 1d), plus 55 non-infected ESKD patient samples. For the Wave 2 cohort (17 patients), transcriptomic data were available for 90 samples collected during acute COVID-19 infection (median of 6 samples per patient, range 3–7), plus 17 pre-infection samples and 12 convalescent samples.

Prior to further analysis, genes with insufficient counts were removed using edgeR's filterByExpr function[81]; for differential expression analyses, the 'group' argument was set to the main group of interest. For all analyses, gene expression was TMM normalised[82], converted to counts per million (CPM) and log-transformed. We primarily used ENSEMBL identifiers[83], however for plots we report the HGNC gene ID[84] where available. For analyses that considered multiple proteins simultaneously (PCA, WGCNA, MEFISTO, supervised learning), we additionally: i) removed genes with low variance (33% of genes with the lowest maximum absolute deviation) using the M3C package[85]; ii) centred and scaled the data.

### Plasma proteomics

We performed proteomics on EDTA plasma samples using the aptamer-based SomaScan platform (Somalogic, Boulder, Colorado, USA). The SomaScan v4.1 assay contains 7288 modified-aptamers (Somamers) that target human proteins. Since more than one aptamer may target the same protein, these 7288 aptamers map to 6347 unique proteins. 48 Somamers were removed due to QC failure, so the final dataset contains 7240 Somamers representing 6323 unique proteins. We annotated these proteins using the Human Protein Atlas[86]; 4980 proteins were labelled as intracellular, 1586 were annotated as membrane proteins and 1160 as secreted (Supplementary Figure 23A). Many proteins were labelled as both intracellular and as membrane or secreted, reflecting the biology of protein storage and extra-cellular secretion/excretion (Supplementary Figure 23B).

We report proteins by their corresponding HGNC gene ID[84], which provides a more standardised nomenclature compared to protein names and allows direct comparison with the transcriptomic data.

Where multiple Somamers related to the same protein, we retained these Somamers for univariate analyses such as differential abundance analyses. However, for analyses that considered multiple proteins simultaneously (PCA, WGCNA, MEFISTO, supervised learning), we selected one Somamer at random to represent each protein. One COVID-19 positive sample in the wave 2 cohort failed QC and was excluded from the analyses. The expression values for each Somamer were inverse-rank normalised prior to downstream analyses.

For the Wave 1 cohort, proteomic data were available for 86 samples from 37 COVID-19 positive ESKD patients (median 3 samples per patient, range 1-3), plus 53 non-infected ESKD patients. For the Wave 2 cohort ($n = 17$ patients), following QC, proteomic data were available for 102 samples collected serially during acute COVID-19 infection (median of 6 samples per patient, range 5–7) and 16 pre-infection samples. For one patient, a pre-infection plasma sample was unavailable.

### Statistics and reproducibility

No statistical method was used to predetermine sample size. Analysts were not blinded to COVID-19 status or severity.

### Differential expression analyses: COVID-19 positive versus negative

We compared COVID-19 positive and negative patients using linear mixed models (LMM), which account for serial samples from the same individual[87]. Age, sex, and ethnicity were included as covariates. A random intercept term was used to estimate the variability between individuals in the study and thus account for repeated measures. We performed differential expression analyses for the transcriptomic data and the proteomic data. The regression model for these analyses in Wilkinson-style notation was:

$$E \sim covid\_status + sex + age + ethnicity + (1 \mid individual)$$

Where, E represents expression (gene or protein, depending on the data type being analysed) and covid_status was a categorical variable (COVID-19 positive or negative).

For differential expression of proteins, we applied LMM using the lmerTest package[88]. Differential gene expression analysis was performed using the same model formula, applied using the differential expression for repeated measures (dream) pipeline[89] in the variancePartition package[90]. For all data types, we fitted LMM using

restricted maximum likelihood (REML) and calculated *P*-values using a type 3 F-test, in conjunction with Satterthwaite's method for estimating the degrees of freedom for fixed effects[88]. Multiple testing correction was performed using the Benjamini-Hochberg method and a 1% FDR used for the significance threshold.

The Wave 1 cohort was analysed separately to the Wave 2 court. For Wave 1, we compared samples from COVID-19 positive ESKD patients to COVID-19 negative ESKD patients. For Wave 2, we compared samples from COVID-19 positive ESKD patients to samples from these patients taken approximately 8 months prior to infection.

When reporting the number of differentially expressed proteins in the text we refer to the number of unique proteins rather than the number of significant Somamers.

### Testing transcriptomic and proteomic features for association with COVID-19 severity

We performed a within-cases analysis, testing for the association of gene expression with COVID-19 severity at time of sampling. We used the four-level WHO severity rating (mild, moderate, severe, critical), which could vary between samples from the same individual reflecting the clinical status at the time the same was taken. We again used a linear mixed model to account for samples from the same individual. The regression model was:

$$E \sim covid\_severity + sex + age + ethnicity + (1|individual)$$

The covid_severity variable represents severity at the time of the sample and was encoded using orthogonal polynomial contrasts to account for ordinal nature of severity levels.

COVID-19 positive samples from the Wave 1 cohort were analysed separately to those from the Wave 2 cohort.

The same approach was used for the proteomics data.

### Gene set variation analysis

To identify pathways that were up- or down-regulated in COVID-19 positive versus negative samples, we applied gene set variation analysis (GSVA)[30]. To define gene sets, we used the MSigDb C2 canonical pathways[91]; we discarded sets with less than ten genes. We additionally included a gene set for the peripheral immune response defined for patients with severe COVID-19[31] and a set of type 1 interferons active in patients with systemic lupus erythematosus (SLE)[92]. After reduction of genes into gene sets, we then performed testing for dysregulated pathways using the same linear mixed modelling approach as for the differential gene and protein expression analyses. *P*-values were adjusted by Benjamini-Hochberg, with a significance threshold of 1% FDR.

To dissect out the key molecules underpinning enriched pathways, we examined the genes that comprise these pathway terms and identified which of these featured most prominently in the differential gene expression analysis.

We repeated this procedure for testing of association of pathways with severity at the time of sample using the 4-level ordinal score.

We then applied the same approach to the proteomics data for the COVID-19 positive versus negative analysis, and for testing associations with COVID-19 severity at the time of sample.

### Robust rank aggregation

The Wave 1 and Wave 2 cohorts were analysed separately for both the differential expression analyses between COVID-19 positive and negative samples and for the within-cases severity analyses. To identify the associations that were most consistent between the Wave 1 and Wave 2 cohorts, for each analysis, we integrated the *P*-values for each cohort using robust rank aggregation (RRA)[93]. This method identifies features that are ranked higher than expected across multiple lists. RRA generates a significance score analogous to a *P*-value; we -log10

transform these values such that a larger score indicates more consistent associations between the Wave 1 cohort and the Wave 2 cohort. RRA was applied to the results of the transcriptomic, proteomic and GSVA analyses comparing COVID-19 positive versus negative samples from Wave 1 and Wave 2. Similarly, it was applied to the analyses testing for association of molecular features with COVID-19 severity at the time of sampling.

### Modelling modular longitudinal trajectories

We examined the temporal trajectories of the transcriptome following infection, by explicitly modelling molecular markers with respect to time following COVID-19 symptom onset. We used a two-step approach.

Step 1. To aid biological interpretation, we first applied a dimension reduction strategy using weighted gene correlation network analysis (WGCNA)[34] to identify modules of correlated molecular features. For this analysis, we combined samples from the Wave 1 and Wave 2 cohorts. Additionally, since our goal was to perform longitudinal analysis, we only selected patients who had been sampled at least three times prior to 21 days following COVID-19 symptom onset. The default implementation of WGCNA is not designed for use with non-independent samples[94], so we modified the analysis pipeline by generating a correlation matrix using a repeated measures correlation metric (rmcorr) that is appropriate for repeated measures[95]. We used WGCNA's pickSoftThreshold from similarity function to pick the minimum soft-thresholding power that satisfied the minimum scale free topology fitting index ($R^2 > 0.85$) and maximum mean connectivity (100). We subsequently defined signed adjacency and topological overlap matrices before applying average-linkage hierarchical clustering. We cut this tree with a hybrid dynamic tree cutting algorithm, with the parameters deepSplit = 4 and minClusterSize = 30[96]. Finally, we defined eigengenes for each module and merged those with a distance less than 0.25. The eigen-genes provide a numerical representation for each module of co-expressed genes.

We used the same approach to analyse the proteomic data.

Step 2. To examine the trajectory of each module over time, we fitted a linear mixed model with time from symptom onset as an independent variable and the eigengene (or eigenprotein in the case of proteomic modules) as the dependent variable. Time was defined for each sample as time from first symptoms; where date of first symptoms was not available, we instead used date of first positive swab. Samples that were taken more than 21 days from each individual's baseline date were excluded. We used R's bs function to fit a polynomial spline of degree two to model the expression of modules with respect to time from baseline[97]. To test whether modules displayed different temporal patterns according to the overall clinical course of COVID-19 (defined as a binary variable indicating whether the peak WHO severity score was mild/moderate or severe/critical), we included clinical course as a covariate in the model, and an interaction term between time from symptom onset and clinical course (TxCC).

The regression model used is displayed using Wilkinson-style notation below.

$$eigenexpression \sim clinical\_course * time + sex + age + ethnicity + wave + (1|individual)$$

We extracted the *P*-values for the TxCC term in this model and applied Benjamini–Hochberg adjustment, using 5% FDR as the significance threshold. A significant interaction effect for the TxCC term indicates that the module has a different temporal profile in mild/moderate versus severe/critical disease.

### Additional WGCNA module annotation and association testing

To better understand the biological information reflected in the transcriptomic and proteomic modules, we further characterised them

through a multi-pronged analytical strategy. We tested association of eigen-genes and eigen-proteins with other variables. First, we tested for the association of the modules with WHO severity at the time of the sample using the LMM approach described above in subsection Testing transcriptomic and proteomic features for association with COVID-19 severity, i.e.:

$$E \sim covid\_severity + sex + age + ethnicity + wave + (1|individual)$$

Second, since PBMCs represent a mixed population of immune cells, we investigated whether disease trajectory-associated transcriptomic modules might reflect shift in cell type proportions. To this end, we applied CIBERSORTx, a computational algorithm to impute immune cell fractions from RNA-seq data (see subsection Cell fraction imputation below). We then tested for correlations between these imputed immune cell proportions and module eigengenes using LMM:

$$eigenexpression \sim cell\_fraction + sex + age + ethnicity + wave + (1|individual)$$

Both these models included an additional fixed effect ('wave') to reflect the cohort.

Third, we performed pathway enrichment analysis on the modules using the R package clusterProfiler's 'enricher' function[98]. Gene sets were defined using MSigDB C2 canonical pathways[91].

Lastly, to understand the relationship between the transcriptomic and proteomic modules, we performed correlation analysis using LMMs.

5% FDR was used for statistical significance for these analyses.

### Cell fraction imputation
We used CIBERSORTx[99] to impute cell fractions from the normalised bulk RNA-seq dataset. The program was run with default parameters We inferred the cell fractions of 22 immune cell types in the isolated PBMCs of each sample using the LM22 signature matrix file[100].

### Multi-omic longitudinal factor analysis with MEFISTO
MEFISTO[101] is an extension of Multi-Omics Factor Analysis (MOFA) that can exploit temporal relationships between samples to find factors that change over time (from baseline). We used this method to find joint factors of variation in the transcriptomic and proteomic datasets. For the MEFISTO analysis, we used the same set of samples as in the network analysis and applied the same pre-processing steps to the data (see Methods−network analysis). Additionally, we removed genes with the lowest maximum absolute deviation[85] such that the number of genes retained were equal to the number of unique proteins measured (6,323) to avoid imbalance numbers of features between the transcriptomic and proteomic data which can impact the MEFISTO algorithm. Using the 'slow' convergence criterion, MEFISTO identified 8 factors that had a minimal variance explained of 1% in at least one data modality.

We then applied the longitudinal model described earlier to test for an interaction effect between time from first symptoms and clinical course, with a latent factor identified by MEFISTO as the dependent variable. The regression model used is displayed using Wilkinson-style notation below:

$$latent\_factor \sim clinical\_course * time + sex + age + ethnicity + wave + (1|individual)$$

### Longitudinal modelling of cytokines and cytokine receptors
We modelled the temporal profiles of 232 plasma proteins that fell within the KEGG pathway 'Cytokine-cytokine receptor interaction'. As for the longitudinal analyses described earlier, we used a linear mixed model with a time x clinical course interaction term.

$$P \sim clinical\_course * time + sex + age + ethnicity + wave + (1|individual)$$

$P$ values for the time x clinical course interaction were extracted and adjusted for multiple testing with the Benjamini-Hochberg procedure, with significance threshold of 5% FDR.

### Supervised learning
The goal of this analysis was to predict clinical severity from the molecular features (transcriptomic, proteomic or both). We performed supervised learning using the R caret framework[102]; caret uses the randomForest package to fit random forest models and glmnet[103] to fit lasso models. For this analysis, we only included samples on which both transcriptomics and proteomics had been performed. We then selected the earliest sample for each individual at which they had reached their peak COVID-19 WHO severity score, so that there was one sample per patient. We then categorised the clinical severity score corresponding to each sample into a binary variable such that patients with a WHO severity score of mild or moderate were considered mild/moderate and those with a WHO score of severe or critical were considered severe/critical. This resulted in $n = 37$ mild/moderate samples and $n = 14$ severe/critical samples.

We trained models using Monte Carlo cross-validation for: (i) the plasma proteomic data alone (6323 features); (ii) the PBMC RNA-seq data alone (12,225 features); and (iii) the combined proteomic and RNA-seq datasets. The first step in this training process was to create 200 random partitions of the data, such that 80% of the data was used to train the model in each resample and 20% was retained as a validation set. In each resample, we calculated the area under the curve (AUC) of the receiver operating characteristic (ROC) curve. We then calculated confidence intervals for the 200 AUC-ROC values generated for each model and feature type.

The random forest model's parameters were kept constant at 500 trees and the mtry value (number of proteins randomly sampled as candidates at each node) was calculated as the square root of the number of features. After cross-validation, we fitted a final random forest model using the entirety of the dataset. We extracted important features from this model using the R randomForestExplainer package, based on the accuracy decrease metric (the average decrease in prediction accuracy upon swapping out a feature). For the lasso model, the lambda value that maximised the mean AUC-ROC during cross-validation was selected. We recorded the features selected by the lasso model in each data resample; feature importance was subsequently defined as the number of models in which each feature had a non-zero coefficient. The feature importance metrics from both models were scaled by dividing their values by the maximum value, such that the most important feature has an importance metric of 1.

### Differential gene expression analysis: pre-infection versus recovery samples
For the 12 individuals in the Wave 2 cohort for whom we collected a convalescent sample (approximately 2 months post-infection; range 41-55 days from the initial sample), we performed a differential gene expression analysis comparing these samples to the paired pre-infection samples using LMM, implemented with the R dream package[89]. Age, sex and ethnicity were included as covariates and a random intercept term used to account for the paired nature of the samples. Statistical significance was defined as 1% FDR. To identify enriched pathways in the list of differentially expressed genes, we performed overrepresentation analyses using the same approach as described above for annotating the WGCNA modules.

## Flow cytometry

Flow cytometry analysis was performed on a subset of the Wave 2 PBMC samples. We examined samples taken during acute COVID-19 from 17 patients (of whom 9 patients had a mild/moderate clinical course and 8 patients with severe/critical course), and pre-infection samples from 15 of these same patients. 12 samples with low cell number recovery (less than 10,000 PBMCs) were excluded from the analysis.

Cryopreserved PBMCs were thawed in humidified 37 °C, 5% $CO_2$ incubator and resuspended in thawing medium (RPMI, 20% FBS). PBMCs were washed twice with PBS and stained with Zombie Yellow LIVE/DEAD (Biolegend) following the manfacturer's protocol to exclude dead cells. Then, PBMCs were washed twice with FACS buffer (1% BSA, 0.09% Azide, 1 mM EDTA), and Fc receptors were blocked with Human TruStain Fc Receptor Blocking Solution (Biolegend). Then, surface staining were performed using the selected fluorochrome-conjugated monoclonal antibodies detailed in Supplementary Table 5 for 20 min at 4 °C. Following incubation, cells were fixed and permeabilized using the eBioscience™ Foxp3/Transcription Factor Staining Buffer Set (Invitrogen) for intracellular staining. Cells were incubated with selected antibodies or isotype controls for 30 min at 4 °C and resuspended in FACs buffer for analysis. Aurora Spectral Flow Cytometry (Cytek®) and FlowJo software, version 10 (Tree Star Inc. Ashland, OR, USA) were used for analysis of all samples. The gating strategy used for flow cytometry is shown in Supplementary Figs. 27-28. Prior to gating cell population of interest, cell debris was removed based on FCS/SSC and only live cell (BV570 Zombie Yellow - negative) populations were analysed.

## Flow cytometry statistical analysis

Flow cytometry statistical analysis was performed with GraphPad Prism (v9). To evaluate decomposition performance by CIBERSORTx analysis, cell proportion estimates were compared to cell percentages from Flow Cytometry analysis using Pearson's correlation analysis ($n = 68$ samples). We were unable to examine for the presence of LDGs using our flow cytometry data since this was performed on cryopreserved PBMCs and LDGs do not survive the freeze-thaw process (whereas we performed transcriptomics on RNA extracted from fresh PBMCs). We observed significant correlation of estimated cell proportions from CIBERSORTx analysis compared to proportions measured by flow cytometry for all other cell types (Pearson $r > 0.4045$, two-tailed $p$-value $< 0.0001$).

For severity analysis, one sample per patient was selected at a time that coincided with the expected spike in the inflammatory response (nearest sample to day 7 after symptom onset; no more than ±72 h). Patients were classified according to the overall peak illness severity into two groups (mild/moderate = 9, severe/critical = 8). Change of cell proportion across time were accessed by grouping samples into 4 days interval post COVID-19-positive test. One-way ANOVA was used to calculate significant differences between multiple groups with Dunnet's correction for multiple-way comparisons. Significance is based upon $p$-value $< 0.05$.

## Reporting summary

Further information on research design is available in the Nature Portfolio Reporting Summary linked to this article.

## Data availability

The individual-level transcriptomics (counts), proteomics and flow cytometry data are available without restriction from Zenodo (https://doi.org/10.5281/zenodo.6497251). Processed subsets of these data corresponding to specific Figures are provided in the Source Data file. The raw RNA-seq reads are under restricted access to comply with UK GDPR legislation and have been deposited in the European Phenome-Genome archive (EGA) under study accession EGAS00001006778; requests for access can be made to the Data Access Committee.

In this study, we utilised the whole blood bulk RNA-seq generated by the COvid-19 Multi-omics Blood ATlas (COMBAT) Consortium study[7], which is available from Zenodo (https://doi.org/10.5281/zenodo.6120249). We also used the SomaScan proteomics data of Filbin et al.[12], accessed from Mendeley Data (https://doi.org/10.17632/nf853r8xsj.2). Source data are provided with this paper.

## Code availability

An archived GitHub repository containing custom analysis code is available from Zenodo (https://doi.org/10.5281/zenodo.7333789).

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

## Acknowledgements

This research was partly funded by Community Jameel and the Imperial President's Excellence Fund and by a UKRI-DHSC COVID-19 Rapid Response Rolling Call (MR/V027638/1) (to J.E.P.), and by funding from UKRI/NIHR through the UK Coronavirus Immunology Consortium (UK-CIC) (to M.B.). We also acknowledge the National Institute for Health Research (NIHR) Biomedical Research Centre based at Imperial College Healthcare NHS Trust and Imperial College London. The views expressed are those of the author(s) and not necessarily those of the NHS, the NIHR or the Department of Health. J.E.P. was supported by UKRI Innovation Fellowship at Health Data Research UK (MR/S004068/2). D.C.T. is

supported by a Stage 2 Wellcome-Beit Prize Clinical Research Career Development Fellowship (20661206617/A/17/Z and 206617/A/17/A) and the Sidharth Burman endowment. M.C.P. is a Wellcome Trust Senior Fellow in Clinical Science (212252/Z/18/Z). N.M.-T. and E.S. are supported by Wellcome Trust and Imperial College London Research Fellowships, and CLC by an Auchi Clinical Research Fellowship. The funders had no role in study design, data collection and interpretation, or the decision to submit the work for publication. We thank the patients who volunteered for this study and the staff at Imperial College Healthcare NHS Trust (the Imperial College Healthcare NHS Trust renal COVID-19 group and dialysis staff): Appelbe M, Ashby DR, Brown EA, Cairns T, Charif R, Condon M, Corbett RW, Duncan N, Edwards C, Frankel A, Griffith M, Harris S, Hill P, Kousios A, Levy JB, Loucaidou M, Lightstone L, Liu L, Lucisano G, Lynch K, Mclean A, Moabi D, Muthusamy A, Nevin M, Palmer A, Parsons D, Prout V, Salisbury E, Smith C, Tam F, Tanna A, Tansey K, Tomlinson J, Webster P. We also acknowledge the efforts of renal specialist doctors in training for assistance with recruiting patients to this study. We thank Dr Arnav Mehta and Dr Michael Filbin, Massachusetts General Hospital and Broad Institute, Cambridge, MA, USA, for their help enabling us to re-analyse the data from Filbin et al.[12]. We acknowledge the Imperial College Research Computing Service (https://doi.org/10.14469/hpc/2232).

## Author contributions

J.S.G.: transcriptomic and proteomic data analyses (primary analyst), wrote the paper (primary draft), visualisation. N.B.B.: sample processing, flow cytometry analysis, study logistics, transcriptomic data analysis, wrote the paper (review and editing), visualisation (flow cytometry). A.P.: transcriptomic and proteomic data analyses. C.L.C.: study logistics, patient recruitment and sample collection, clinical phenotyping. T.H.M.: sample processing. N.M-T.: study logistics, patient recruitment and sample collection, clinical phenotyping. D.P.: flow cytometry analysis. P.M.M.: sample processing. S.L.: study logistics, patient recruitment and sample collection, sample processing. E.S.: patient recruitment and sample collection. S.P.M.: patient recruitment and sample collection. M.F.P.: patient recruitment and sample collection. M.W.: conceived the study, study logistics, obtained ethical approval, led patient recruitment and sample collection, funding acquisition. M.C.P.: conceived the study, study logistics, clinical phenotyping, wrote the paper (review and editing), funding acquisition. M.B.: conceived the study, study logistics, supervised sample processing, supervised flow cytometry analysis, wrote the paper (review and editing), funding acquisition. D.C.T.: conceived the study, study logistics, patient recruitment and sample processing, supervised flow cytometry analysis, wrote the paper (primary draft), funding acquisition. J.E.P.: conceived the study, study logistics, clinical phenotyping, supervised the transcriptomic and proteomic analyses, overall project supervision, wrote the paper (primary draft), funding acquisition. All authors critically reviewed and approved the manuscript before submission.

## Competing interests

None of the authors have any patents (planned, pending or issued) or competing interests relevant to this work. Other interests unrelated to this work: S.P.M. reports personal fees from Celltrion, Rigel, GSK and Cello; M.C.P. reports consulting honoraria with Alexion, Apellis, Achillion, Novartis and Gyroscope; D.C.T. reports speaker and consultancy fees from Astra-Zeneca and Novartis; J.E.P. has received travel and accommodation expenses and hospitality from Olink proteomics to speak at Olink-sponsored academic meetings (none within the past 5 years). None of the other authors have any interests to declare.
