## [Peer Review File · Nature Communications]

REVIEWER COMMENTS

Reviewer #1 (Remarks to the Author):

1. What are the noteworthy results?

In summary, the authors demonstrate dynamic transcriptomic, proteomic, and cellular signatures that vary both with time and COVID-19 severity. They show that in patients with a severe clinical course there is increased type 1 interferon signaling early in the illness, with increases in pro-588 inflammatory cytokines later in the disease. They identify plasma levels of the proposed alternative 589 SARS-CoV-2 receptors, LRRC15, as the strongest predictor of COVID-19 severity and show that immune cells display dysregulated gene expression two months following COVID-19, with upregulation of clotting-related genes.

Will the work be of significance to the field and related fields? How does it compare to the established literature? If the work is not original, please provide relevant references.

Response. Patients with ESKD are at high risk of worse outcomes from COVID. This study dives into why, using multi-omics. Despite a small sample size, they do find a significant result. Additionally, they recapitulate the finding that patients with ESKD have more thrombosis

Does the work support the conclusions and claims, or is additional evidence needed?

Response. The work supports the conclusions

Are there any flaws in the data analysis, interpretation and conclusions? Do these prohibit publication or require revision?

Response.

Few major comments

1. While adjusting for association between proteomics and outcome in ESKD patients, why weren't more clinical covariates (eg. duration of time spent on hemodialysis) adjusted for

2. Was collinearity checked between the plasma proteome

3. Were ensemble methods used for prediction of prot-gene or prot-prot interactions.

4. How was clotting outcome assessed.

These need clarification and revisions

Is the methodology sound? Does the work meet the expected standards in your field?

Response. Yes but there was no independent validation cohort.

Is there enough detail provided in the methods for the work to be reproduced?

Response. Unable to assess.

Reviewer #2 (Remarks to the Author):

Using plasma proteomics, and RNA-sequencing and flow cytometry of immune cells, Gisby et al identified multiomic and temporal signatures of COVID-19 severity in end-stage kidney disease (ESKD) patients. The author further identified LRRC15 as biomarker for clinical outcome and reported convalescent patients still display dysregulated gene expression related to vascular, platelet and

coagulation pathways. Overall, this study is different from many other published works on COVID-19 multi-omics because the study subjects are uniquely focused on ESKD patients with COVID-19. However, most of the analysis did not differentiate what's unique/novel for their ESKD patient cohorts that are different than other published results on non-ESKD patients (re-analyze published data). As a result, a major opportunity to sell the novelty of this study is lost. Meanwhile, I spotted some technical issues, unclearness, and the omissions of a few relevant references, which will also need to be addressed. Overall, I believe most of my points can be still addressed in a major revision and I would like to see a revised manuscript with the following points addressed.

1. This is probably my biggest suggestion. As the author are probably aware, there are already tones of literatures on COVID-19 plasma proteomics (e.g. PMID: 33969320, 32717743, etc.) or even in conjunction with metabolomics (e.g. PMID: 32492406, 34489601), and PBMC immune cell transcriptome even with single-cell resolution (e.g. PMID: 33171100, 35216672), and how those correlated with COVID-19 severity in non-ESKD patient cohorts. This study has a great potential to be different from many of these previous publications and adding more values to the field of COVID-19 research because it is uniquely focused on ESKD patients with COVID-19. However, this reviewer was disappointed to see that such a great opportunity of novelty is not fully utilized in the current analysis. Almost all of the current analyses are done purely by correlating severity to different multiomic analytes or modules, without a clear distinction of which findings are unique to ESKD-COVID-19 patients that are not present in non-ESKD COVID-19 patients. There are already a lot of published datasets of plasma multi-omics (e.g. PMID: 33969320, 32717743, 33171100, 35216672) and PBMC transcriptomic on non-ESKD-COVID-19 patients (e.g. PMID: 33171100, 35216672, 33879890, 33657410) across different severity versus healthy controls. Can the authors perform similar analysis on some of these already published dataset on non-ESKD COVID-19 patients and compare with the ESKD COVID-19 analysis results? From there, the author can specifically emphasize which findings may be unique for ESKD-COVID-19 patient cohorts that are not present in non-ESKD Covid-19 patients. This reviewer thinks those comparisons will help make this paper a true valuable addition to the field of COVID-19 research.

2. Comparison of pro-thrombotic signals in non-ESKD COVID-19 patients at convalescence and potential connections with other long-covid symptoms. One of the most interesting findings the authors report is the enrichment of "Platelet activation..." signatures in convalescent stage of their cohort. This is interesting but the reviewer is wondering whether this may be something specific for ESKD-COVID-19 patient cohorts or not. It will be important to check if all of these signatures are also enriched in non-ESKD COVID-19 patients at convalescent stage. Even if some of the signatures are not unique for ESKD-COVID19, it is still worthwhile to report. There are already published datasets on convalescent non-ESKD COVID-19 patients (e.g. PMID: 35216672, 33657410, etc.) that the authors can easily access to perform similar analysis to address this important question. In addition, some of the published dataset also contains patient-reported long-covid symptoms. Will any of the signatures found here are also associated with certain long-COVID symptom in those published datasets? If so, the finding here will be even more interesting to a much broader audience.

3. Lack of key reference citations that reported similar/relevant findings. Some of the findings reported in this manuscript are actually consistent with previous literatures on non-ESKD COVID-19 cohorts. It is inappropriate to omit to cite the previous publications even if those were performed on non-ESKD COVID-19. This reviewer only provided a few examples below, but the authors will need to address this systematically beyond these examples.

3a. Page8, line 269-273, the authors state "...higher lymphocyte related gene signatures and lower myeloid-related ones is a favourable prognostic sign...". These findings are actually consistent with many published works on non-ESKD COVID-19 cohort that decreased proportion of lymphoid cells and increase of myeloid cells are associated increase of disease severity (e.g. PMID: 32669297, 33171100).

3b. Page9, line 291-293, the authors found "...severe/critical patients, there was a progressive drop in the proportion of non-classical monocytes..." This finding is indeed consistent with many published works on non-ESKD COVID-19 cohort that reported the decrease of non-classical monocytes is associated with severity of COVID-19. (e.g. PMID: 32514174, 33171100)

3c. In the result section "Longitudinal cytokine/chemokine analysis reveals distinct temporal profiles

that distinguish disease severity”, the authors reported that different plasma proteins exhibited distinct temporal patterns between severe/critical versus mild/moderate ones. The reviewer is wondering do any of these markers are consistent with (or different from) previous published data on similar plasma omic analysis of non-ESKD COVID-19 cohort (e.g. PMID: 33969320, 32717743, etc.)

4. Opportunity to synergize plasma proteomics with PBMC immune cell transcriptome. The authors have performed a lot of analysis separately on plasma proteomics with PBMC immune cell transcriptome, but it will be important to synergize the two, otherwise the value of measuring both in the same cohort will be compromised. The authors did try to do the machine learning with the two datasets together, but the conclusion that the prediction proteome along works even greater than the multi-omic data is not an appropriate stopping point and can be misleading. Is there any other analysis that the authors can do to sell more on the synergy of the two datasets? For example, does the combination of two datasets help separate the patients with different severity better than only using one dataset (e.g. PMID: 34489601, Lee et al showed that combining plasma omic and PBMC transcriptome can classify patients with different severity even better than just plasma data.). Or, does the enrichment results of the plasma omics independently matches the results of immune transcriptomic and thus enhance the message of what biological process are going on in those patients? (e.g. the monocyte transcriptomic and plasma proteomic integration analysis in PMID: 33171100).

5. Clarification of differential analysis between mild and severe on longitudinal sample where each patients has multiple blood draws taken. It is not clear to this reviewer how do the authors statistically account for multiple sampling of the same individuals, when performing the differential analysis between mild and severe disease, because there could be multiple blood draws taken from the same individual. This should be clarified and justified.

6. The title of the manuscript specified “severity predictor” which can be misleading because the word “prediction” generally implies measure something at earlier time point to predict future events. It seems like this is not the case in the analysis of LRR15. The severity seems to be at the exact same time point of the plasma omic measurement. If that’s the case, the authors should better use words “associated marker” or something like that to replace “predictor” in order to avoid the potential misleading interpretation from readers.

7. Validation cohorts for machine learning model. A validation cohort is generally needed for machine learning related biomarker analysis. This manuscript already have two cohorts, so why not use one as a training set and the other as an independent validation cohort. Otherwise, it may seems to be using the same dataset to train and “predict” which is not as compelling.

8. Is LRR15 also a biomarker for severity in non-ESKD COVID-19 cohort, perhaps by analyzing published plasma omic dataset for non-ESKD COVID-19 patient cohort?

9. Some of the figures may need more clarifications. For example, in Figure 1A, this reviewer is confused regarding what does “+ve” mean for the Wave1 cohort, why Wave 2 cohort does not have “+ve”?

"Multi-omics identify LRRC15 as a COVID-19 severity predictor and persistent pro-thrombotic signals in convalescence" by Gisby *et al.* Reference NCOMMS-22-15678

Response to reviewers

We would like to thank the reviewers for their constructive comments and suggestions regarding our manuscript. We have addressed their specific points, including adding new analyses where appropriate, as detailed below. Reviewer comments are in ***blue italics***.

We report the line numbers of changes to the text. Where we quote passages of the revised text in this response we show changes in **purple**.

In the revised manuscript and Supplementary material changes are highlighted in purple to allow easy identification of changes.

Reviewer 1

Journal: *'What are the noteworthy results?'*

Reviewer: *In summary, the authors demonstrate dynamic transcriptomic, proteomic, and cellular signatures that vary both with time and COVID-19 severity. They show that in patients with a severe clinical course there is increased type 1 interferon signaling early in the illness, with increases in pro-inflammatory cytokines later in the disease. They identify plasma levels of the proposed alternative SARS-CoV-2 receptors, LRRC15, as the strongest predictor of COVID-19 severity and show that immune cells display dysregulated gene expression two months following COVID-19, with upregulation of clotting-related genes.*

Journal: *'Will the work be of significance to the field and related fields? How does it compare to the established literature? If the work is not original, please provide relevant references.'*

Reviewer: *Patients with ESKD are at high risk of worse outcomes from COVID. This study dives into why, using multi-omics. Despite a small sample size, they do find a significant result. Additionally, they recapitulate the finding that patients with ESKD have more thrombosis*

Journal: *Does the work support the conclusions and claims, or is additional evidence needed?*

Reviewer: *The work supports the conclusions*

Journal: *'Are there any flaws in the data analysis, interpretation and conclusions? Do these prohibit publication or require revision?'*

Reviewer:

R1 point 1. *While adjusting for association between proteomics and outcome in ESKD patients, why weren't more clinical covariates (eg. duration of time spent on hemodialysis) adjusted for?*

Author response:

We selected the covariates used in our regression models following evaluation of potential confounders of the associations between proteins and case/control status or severity. We did not show these exploratory analyses in the original manuscript due to space constraints.

Cause of ESKD, presence of diabetes and time since first initiation of haemodialysis did not materially impact the proteome and thus were not included as covariates in the model (for example, see new **Supplementary Figure 24**). To address the reviewer's point and to ensure that these covariates were not confounding the analyses, we performed new sensitivity analyses comparing the results from the existing model (adjusting for age, sex and ethnicity) to models including additional clinical covariates. Specifically, we explored the effects of adjusting for the following additional covariates: cause of ESKD, diabetes, and duration of time since first commencing haemodialysis.

We performed these sensitivity analyses for a) the Wave 1 differential protein abundance analyses between COVID-19 positive versus negative samples, and b) the testing of associations between proteins and COVID-19 severity at the time of the blood sample. We found that the betas (model coefficients) were highly consistent between the models (**Supplementary Figure 2**). We have added a brief summary of this new sensitivity analysis in the manuscript on lines 128-131, with more detailed description in the **Supplementary Material** (page 2).

We also wish to highlight that all samples were taken immediately prior to commencing a haemodialysis session, so potential short-term fluctuations in protein concentrations related to a haemodialysis session should be controlled for.

For the Wave 2 analysis COVID-19 positive versus negative differential protein abundance, the issue of potential confounding by clinical covariates is much less of a concern, since we compared COVID-19 samples to pre-infected from the same individuals (i.e. inter-individual variability is inherently controlled for).

R1 point 2. Was collinearity checked between the plasma proteome?

Author response

There are plasma proteins which have high correlation with one another. As the reviewer implies, collinearity (i.e. complete or very high correlation) between two predictor variables creates problems for a regression model. However, this does not create any problems or issues for the various analyses performed here. We address each of the various analyses that we performed in turn to show why this is so.

i) Regression analyses. Collinearity between proteins is not an issue for our regression analyses (linear mixed models) as we fit a separate model for each protein, and then perform the appropriate p-value adjustment for multiple testing. Thus, we never have two proteins in the same regression model. In fact, in keeping with the standard approach for differential expression analyses we use the protein as the dependent variable, with either case/control status or clinical severity as the independent variables (depending on the analysis), plus covariates (age, sex, ethnicity).

In R notation, the model for the COVID-19 positive versus negative differential expression analyses is:

$E \sim \text{covid_status} + \text{sex} + \text{age} + \text{ethnicity} + (1 \mid \text{individual})$

where, E represents expression and "covid_status" was a categorical variable (positive/negative): see Methods lines 832-860. Thus, the beta coefficient for the

covid_status term represents the difference in expression levels between cases and controls.

For the analysis of proteins associated with severity within the cases, the model was:

$E \sim \text{covid_severity} + \text{sex} + \text{age} + \text{ethnicity} + (1 | \text{individual})$

ii) Network/modular analyses (WGCNA). WGCNA deliberately exploits the correlation structure in the data to our advantage. WGCNA groups correlated features (i.e. proteins in the case of the proteomic data, or genes in the case of the transcriptomic data) into 'modules'. This enables dimension reduction of a very high number of individual molecular features into a smaller number of modules, which can then be tested for enrichment of pathway terms, providing improved biological interpretability.

iii) Supervised learning (random forests and Lasso).

In contrast to the linear mixed model regressions (which analysed each protein one-by-one), the supervised learning analyses model multiple molecular features simultaneously. Indeed, the reason we performed these analyses is that simultaneously modelling the entire proteome or transcriptome has the potential to offer additional insights through capturing non-linear effects or non-additive interactions. Both random forests and Lasso methods are able to cope with highly correlated or collinear features. The random forests algorithm has no assumptions regarding collinearity and is not based on linear regression. Lasso performs variable selection to result in a sparse model through L1 regularisation. Given two or more highly correlated features, it will select one feature that captures the predictive information and shrink the coefficients of the other features to zero.

iv) Principal components analysis and multi-omics factor analysis. These methods are dimensionality reduction methods that reduce correlated features to a smaller set of uncorrelated factors, and so are well suited to the situation of correlated features.

R1 point 3. Were ensemble methods used for prediction of prot-gene or prot-prot interactions?

Author response

For the supervised learning analysis, we used the random forests algorithm, which is considered an ensemble method because it uses a set of many decision trees to 'vote' on the final prediction. We did not, however, create an ensemble method that allowed multiple separate models to vote on the predictions.

In response to the reviewer's suggestion, we performed a new supervised learning analysis that ensembles four models (to predict severity from the transcriptome and proteome, using lasso and random forests). For this new analysis, we trained the models on the Wave 1 cohort and tested them on the Wave 2 cohort. This led to the creation of four models, as a result of using both random forests and lasso to predict severity for each of the transcriptomic and proteomic data modalities. We ensembled these four models for the prediction of severity, finding that the final ensemble model had good performance in the Wave 2 cohort (AUC = 88.6%) despite the small size of the training set. We do not feel this substantially changes the conclusions or message of our study, but we have included this in the Supplementary material.

R1 point 4. How was clotting outcome assessed?

Author response

The reviewer may have misunderstood the reported findings. To clarify, we performed a differential gene expression analysis comparing the PBMC transcriptome of convalescent samples taken ~2 months after COVID-19 to pre-infection samples from the same individuals. This revealed 25 differentially expressed genes (**Table 1**). We performed pathway enrichment analysis on this list of genes, which revealed that these dysregulated genes were enriched for pathways including “Platelet activation, signaling and aggregation” and “Formation of fibrin clot/clotting cascade”. The pathway enrichment was performed using gene annotations from the MSigDb C2 canonical pathways. To clarify, the outcome here is not clinical events (e.g. deep vein thrombosis or pulmonary embolus); our sample size is too small to meaningfully assess risk of clinical thrombotic events but population-based studies have demonstrated elevated risk of clotting for a prolonged period after COVID-19 (see Discussion lines 637-673). Our data provide a potential mechanism that may contribute to this pro-thrombotic tendency.

To improve clarity, we have revised the Discussion as follows (changes in purple):

“Leveraging this, we demonstrate that there is chronic activation of **gene expression related to** vascular, platelet and coagulation pathways for a prolonged period after clinical resolution of disease.”

R1 point 5

Journal: “Is the methodology sound? Does the work meet the expected standards in your field?”

Reviewer: Yes but there was no independent validation cohort.

Author response

Regarding the issue of a validation cohort, we agree that since all the patients are from a single centre, we do not have a truly independent validation cohort. However, for the majority of the analyses, we analysed the Wave 1 cohort separately from the Wave 2 cohort, and report the findings that are consistent across both cohorts using robust rank aggregation (RRA). We also showed that the results from Wave 1 are similar to those from Wave 2 (**Supplementary Figure 2**). Thus, for the proteins and genes differentially expressed between COVID-19 positive and negative patients, and for the proteins and genes associated with COVID-19 severity, we have validation in independent sets of patients, albeit from the same hospital.

For the supervised learning analyses, the situation is different in that we analysed samples from Wave 1 and Wave 2 cohorts together since a large sample size is required for effective supervised learning. We discuss this further in our response to **Reviewer 2 Point 7**.

R1 point 6

Journal: “Is there enough detail provided in the methods for the work to be reproduced?”

Reviewer: Unable to assess.

Author response

As well as the details in the Methods section, we also provide the individual-level proteomic and transcriptomic data. This available without restriction at the open-access Zenodo repository: [doi: 10.5281/zenodo.6497251](https://doi.org/10.5281/zenodo.6497251)

We also provide all the code used in the analysis at <https://github.com/jackgisby/covid-longitudinal-multi-omics>

Together, this combination of data and code allows any researcher to reproduce our findings. We consider this to be in keeping with best practice in transparent/reproducible research. The data and code availability statements are after the Methods section.

Reviewer 2

Using plasma proteomics, and RNA-sequencing and flow cytometry of immune cells, Gisby et al identified multiomic and temporal signatures of COVID-19 severity in end-stage kidney disease (ESKD) patients. The author further identified LRRC15 as biomarker for clinical outcome and reported convalescent patients still display dysregulated gene expression related to vascular, platelet and coagulation pathways. Overall, this study is different from many other published works on COVID-19 multi-omics because the study subjects are uniquely focused on ESKD patients with COVID-19. However, most of the analysis did not differentiate what's unique/novel for their ESKD patient cohorts that are different than other published results on non-ESKD patients (re-analyze published data). As a result, a major opportunity to sell the novelty of this study is lost. Meanwhile, I spotted some technical issues, unclearness, and the omissions of a few relevant references, which will also need to be addressed. Overall, I believe most of my points can be still addressed in a major revision and I would like to see a revised manuscript with the following points addressed.

R2 point 1. *This is probably my biggest suggestion. As the author are probably aware, there are already tones of literatures on COVID-19 plasma proteomics (e.g. PMID: 33969320, 32717743, etc.) or even in conjunction with metabolomics (e.g. PMID: 32492406, 34489601), and PBMC immune cell transcriptome even with single-cell resolution (e.g. PMID: 33171100, 35216672), and how those correlated with COVID-19 severity in non-ESKD patient cohorts. This study has a great potential to be different from many of these previous publications and adding more values to the field of COVID-19 research because it is uniquely focused on ESKD patients with COVID-19. However, this reviewer was disappointed to see that such a great opportunity of novelty is not fully utilized in the current analysis. Almost all of the current analyses are done purely by correlating severity to different multiomic analytes or modules, without a clear distinction of which findings are unique to ESKD-COVID-19 patients that are not present in non-ESKD COVID-19 patients. There are already a lot of published datasets of plasma multi-omics (e.g. PMID: 33969320, 32717743, 33171100, 35216672) and PBMC transcriptomic on non-ESKD-COVID-19 patients (e.g. PMID: 33171100, 35216672, 33879890, 33657410) across different severity versus healthy controls. Can the authors perform similar analysis on some of these already published dataset on non-ESKD COVID-19 patients and compare with the ESKD COVID-19 analysis results? From there, the author can specifically emphasize which findings may be unique for ESKD-COVID-19 patient cohorts that are not present in non-ESKD Covid-19 patients. This reviewer thinks those comparisons will help make this paper a true valuable addition to the field of COVID-19 research.*

Author response

While we agree with the reviewer that delineating whether there are distinct biological processes at work in COVID-19 in ESKD patients is interesting, we feel that the characterisation of our analyses as simply a series of correlations of molecular traits with COVID-19 severity is not a fair representation of our study and misses important aspects of our work. As the reviewer highlights, many studies have done snapshot analysis correlating with severity. In contrast, few have done proper longitudinal analysis (with many studies purporting to have conducted longitudinal analysis in fact having often only a very limited number of timepoints, which are not adequate for construction of molecular trajectories, or even performing cross-sectional analysis on different patients sampled at varying times since symptom onset). In contrast, we had dense serial sampling (median 3 samples per patient for the Wave 1 cohort and 6 for the Wave 2 cohort). Importantly, we had serial samples from **both outpatients and hospitalised patients**, whereas most other studies have been limited to longitudinal analysis of hospitalised patients. One of the reasons we were able to achieve this is that haemodialysis patients must attend medical facility for regular haemodialysis, irrespective of whether they have COVID-19 (whereas the advice for the general population was to isolate at home and only attend hospital in the event of signs of severe disease). The reviewer also appears to have overlooked the fact that uniquely we have pre-infection, acute infection and convalescent samples **from the same individuals**. Another strength of our study is the statistical approach to the analysis of longitudinal data. The analytical approaches to longitudinal data in many previous studies has often been crude (e.g. coarse binning of samples into large time interval windows, followed by statistical tests that do not appropriately account for repeated (i.e. non-independent) measures from the same person). In contrast, we used linear models with a time x severity interaction term to identify proteins and transcripts that have distinct molecular trajectories between mild and severe disease in a statistically rigorous way. **Figure 5** illustrates the value of time-resolved profiles. For example, for IL-11, we observed that plasma IL-11 was lower in severe patients versus mild patients at early timepoints, but showed the reverse pattern later on (**Figure 5D**). This pattern would not be apparent from snapshot measurements or analysis that did not include explicit temporal modelling.

To definitively address the reviewer's question regarding ESKD-specific effects, we would need samples from non-ESKD patients with COVID-19, processed in our centre with data generated on the same platforms (to allow comparison without confounding technical effects). Unfortunately, we did not have access to such samples. The reviewer's suggestion of comparing to other studies can give a general sense of how similar our results are to those in more general patient populations. However, there are important limitations to such inter-study comparisons due to differences in study design, statistical power, assay platforms and other source of non-biological variation, and so we need a degree of caution in claiming ESKD-specific effects based on such comparisons. In addition, patients included in previously studies of severe COVID-19 are often heavily enriched for underlying conditions/comorbidities and unpicking this heterogeneity is challenging. For example, 40/525 (7.5%) of the patients in the INCOV cohort from Su et al paper mentioned by the reviewer (PMID 35216672) had chronic kidney disease, (although the degree of renal impairment was not clear from the available meta-data). Nevertheless, with these caveats in mind, we have attempted to compare our results to other studies.

Transcriptomic comparison

The reviewer references studies using single cell RNA-seq of PBMC (e.g. PMID 33171100). For obvious reasons, these single cell RNA-seq studies do not allow head-to-head comparison with our data (bulk PBMC RNA-seq). We therefore compared our RNA-seq results to the

Covid-19 Multi-omics Blood ATlas (COMBAT) Consortium study (Cell 2022: 185: 916-938, PMID: 35216673), which is also from a UK healthcare setting (thus minimising differences due to variation in clinical practice). The COMBAT study performed bulk RNA-sequencing of whole blood from 77 individuals with COVID-19 and 10 healthy controls. We re-analysed these RNA-seq data so that they were as comparable as possible to our data. Specifically, we used linear mixed models to test the association of genes with COVID-19 status (positive vs. negative) and severity (encoded as an ordinal variable; mild, severe, critical). We also tested the enrichment of gene sets for these comparisons using the GSVA method that we used in our analysis. We then compared the effect estimates effect in our cohorts to those from our re-analysis of the COMBAT data. For the comparison of COVID-19 positive versus negative samples, we found a high degree of concordance between our study and the COMBAT study, both for the gene-level differential expression analysis (COMBAT versus our Wave 1 data $r = 0.68$; COMBAT versus our Wave 2 data also 0.68) and for the GSVA analysis (COMBAT versus Wave 1 $r = 0.54$; COMBAT versus Wave 2 $r = 0.66$) (**Supplementary Figure 17**). Our results for the association of gene expression with COVID-19 severity were also generally consistent with the COMBAT data. For gene-level analysis, Wave 1 $r = 0.56$; Wave 2 $r = 0.59$) and GSVA gene sets (Wave 1 $r = 0.44$; Wave 2 $r = 0.37$) (**Supplementary Figure 18**).

Plasma proteomic comparison

We next sought to compare our plasma proteomic data to those collected in a cohort of non-ESKD patients with COVID-19. We focussed on a study proposed by the reviewer (Filbin *et al.*, PMID: 33969320). We selected this study as it a) had a large sample size, b) because it used SomaScan proteomics, and c) collected samples at multiple timepoints during acute COVID-19, making it highly comparable with our data. We re-analysed the Filbin *et al.* dataset in the same manner as we did for the COMBAT study and compared the results to our study.

The effects estimated for the comparison of COVID-19 cases and controls were positively correlated for both plasma protein abundance (Wave 1 $r = 0.45$; Wave 2 $r = 0.28$) and GSVA protein sets (Wave 1 $r = 0.46$; Wave 2 $r = 0.22$) (**Supplementary Figure 19**). The proteomic associations with contemporaneous severity were also positively correlated for both proteins (Wave 1 $r = 0.20$; Wave 2 $r = 0.55$) and protein sets (Wave 1 $r = 0.22$; Wave 2 $r = 0.39$) (**Supplementary Figure 20**).

In summary, these comparisons reveal that our results are generally similar to those identified in cohorts of COVID-19 patients without ESKD, although our proteomic results are not as concordant with previous studies as the transcriptomic data. This could reflect differences in study design (unlike our controls, those of Filbin *et al.* presented with acute respiratory distress) or technical differences (the study by Filbin *et al.* used an earlier version of the SomaScan platform). However, the lower concordance for plasma proteins than PBMC gene expression could be biological as it is known that circulating proteins are affected by renal failure. **Supplementary File 1U** lists proteomics pathways that were significantly enriched in our data but not in the data of Filbin *et al.*

We found that many of the pathways that we identified in severe COVID-19 in ESKD patients are common to severe COVID-19 in other patient populations. However, further interrogation of our data identified some biologically plausible examples of ESKD-specific effects. For example, one finding that is likely specific to ESKD is the differential temporal dynamics of the erythropoietin receptor (EPOR) in severe/critical COVID-19 versus mild/moderate disease. Erythropoietin (EPO) is a hormone produced by the kidney that promotes red cell formation. ESKD patients fail to produce EPO and consequently require exogenous administration. We observed a waxing and waning profile in severe COVID-19 and a more stable profile in

mild/moderate disease (new **Supplementary Figure 16**). Specifically, in the early part of the illness (first week of symptoms), we observed increased abundance of the EPOR in severe/critical COVID-19 compared to mild/moderate COVID-19. However, later in the disease we observed the inverse pattern, with lower abundance in the severe/critical group.

Revisions to the text: Results lines 351-383, Supplementary Material pages 3-5, Supplementary Figures 16-20, Supplementary File 1U.

***R2 point 2.** Comparison of pro-thrombotic signals in non-ESKD COVID-19 patients at convalescence and potential connections with other long-covid symptoms. One of the most interesting findings the authors report is the enrichment of “Platelet activation...” signatures in convalescent stage of their cohort. This is interesting but the reviewer is wondering whether this may be something specific for ESKD-COVID-19 patient cohorts or not. It will be important to check if all of these signatures are also enriched in non-ESKD COVID-19 patients at convalescent stage. Even if some of the signatures are not unique for ESKD-COVID19, it is still worthwhile to report. There are already published datasets on convalescent non-ESKD COVID-19 patients (e.g. PMID: 35216672, 33657410, etc.) that the authors can easily access to perform similar analysis to address this important question. In addition, some of the published dataset also contains patient-reported long-covid symptoms. Will any of the signatures found here are also associated with certain long-COVID symptom in those published datasets? If so, the finding here will be even more interesting to a much broader audience.*

Author Response

We agree with the reviewer that the presence of gene expression signatures related to platelet activation and coagulation is intriguing, and may link to the observation of prolonged elevation of thrombotic risk even after clinical recovery from COVID-19.

A key and unique aspect of our study is that we had both convalescent samples and paired pre-infection baseline samples on the same patients. We were thus able to control for inter-individual heterogeneity and hidden confounding. We are not aware of any other study with this unique design. As a result, we cannot reliably compare our findings to non-ESKD populations or make confident generalisations outside the context of ESKD (a limitation that we previously highlighted in the Discussion). We note, however, that the elevated risk of clinical clotting events is a generalised feature of COVID-19 that has been observed in population studies (see the papers we cite in the Discussion) i.e. it is not an ESKD-specific effect.

As the reviewer points out, other studies have looked at convalescent samples. However, we emphasise that these studies did not compare convalescent samples with respect to paired pre-infection samples. Rather, they compared to convalescent to control samples from different individuals and are thus their results are subject to many potential confounding factors.

The studies listed above by the reviewer (Su et al Cell 2022 PMID 35216672, Ren et al Cell 2021 PMID 33657410) both involved single cell RNA-seq analysis. Single cell RNA-seq data is not directly comparable to the bulk PBMC RNA-seq data in our study, and thus we cannot make reliable conclusions regarding ESKD-specific signatures based on comparison with these papers. Nevertheless, at the reviewer’s suggestion we re-analysed the data of Ren *et al.* and performed differential gene expression between convalescent samples and healthy controls in monocytes, B cells, CD4 and CD8 T cells. We then compared the 25 differentially expressed genes in our analysis of convalescent samples versus pre-infection samples to the

results from our re-analysis of the data of Ren *et al.* This revealed 17 were significantly differentially expressed (5% FDR) in at least one cell type in the study of Ren *et al.* However, only 5 of these were directionally concordant in our data. Clearly, it is impossible to disentangle what is a technical effect of single cell versus bulk PBMC RNA-seq and biological effects of ESKD. In addition, there may be hidden confounding due to differences in the cases and controls in the study of Ren *et al.*, whereas we had paired samples minimising the effects of inter-individual variation. For these reasons, we feel that it would be scientifically inaccurate to present firm conclusions regarding what is and what is not ESKD-specific from this comparison. We have included a description of this new analysis below in purple titled “*Are the prolonged transcriptomic changes we observed following COVID-19 specific to ESKD patients?*”, but do not believe it adds interpretable information to our study and therefore have not included it in the manuscript.

Regarding the question of ‘long COVID’, we stress that the convalescent samples in our study were from patients who were fully clinically recovered, and not from ‘long COVID’ patients. This is an important and novel aspect of our data i.e. we have found evidence for dysregulation of genes related to clotting pathways in the absence of symptoms. This may tie with epidemiological studies showing elevated pro-thrombotic risk for a prolonged period after infection (see Discussion). Attempting to link our data to long COVID is potentially misleading and risks conflating distinct issues.

Are the prolonged transcriptomic changes we observed following COVID-19 specific to ESKD patients?

Our differential gene expression analysis comparing convalescent samples taken approximately 2 months following infection to pre-infection samples from the same individuals identified 25 differentially expressed genes, many of which relate to clotting pathways. We compared our results to a study by Ren *et al.* that includes samples from healthy controls and individuals recovering from COVID-19. We re-analysed their PBMC single cell RNA-seq dataset (data available from NCBI GEO, accession: GSE158055). After removing convalescent samples whose samples were taken less than 30 days following symptom onset, 38 convalescent samples and 17 healthy control samples remained. After pseudobulking these data, within each cell type we applied a similar normalisation and modelling protocol that we used to identify differences between pre-infection and convalescent timepoints in our own cohort. The statistical modelling approach varied slightly for our data versus our re-analysis of the data of Ren *et al.*, since the pseudobulked data had only one measurement from each individual whereas our study had pre- and post-infection samples from the same individuals. Consequently, we used the limma-voom package to fit linear models on the Ren data rather than the dream package which we used on our data.

We then attempted to compare the 25 genes that were significantly differentially expressed in our data to that of Ren *et al.* Three genes (ENSG00000236304, MT-RNR1 and ENSG00000240093) were not found in their data. Of the remaining 22 genes, 17 were significantly differentially expressed (5% FDR) in at least one cell type in the study of Ren *et al.* However, only 5 of these were directionally concordant in our data (see Figure below).

Clearly, it is impossible to disentangle what is a technical effect of single cell versus bulk PBMC RNA-seq and biological effects of ESKD. In addition, there may be hidden confounding due to differences in the cases and controls in the study of Ren *et al.*, whereas we had paired

pre-infection and convalescent samples minimising the effects of inter-individual variation. For these reasons, we cannot make robust conclusions regarding what is and what is not ESKD-specific from this comparison.

Comparison of healthy controls and convalescent samples in the single cell data of Ren *et al.* Fold changes between convalescent and healthy samples in the single cell RNA-seq data of Ren *et al.* for the genes significantly differentially expressed in our comparison of pre-infection and convalescent samples. The “ESKD cohort” column indicates fold change in the Wave 2 cohort. Colour represents log fold change, with red representing up-regulation in convalescence and blue representing down-regulation. Cells with missing values are grey, indicating that no comparison was made due to low gene counts within the cell type; only cell types with >50% non-missing comparisons are shown. Asterisks denote Benjamini-Hochberg adjusted P-values of less than 0.05.

R2 point 3. Lack of key reference citations that reported similar/relevant findings. Some of the findings reported in this manuscript are actually consistent with previous literatures on non-

ESKD COVID-19 cohorts. It is inappropriate to omit to cite the previous publications even if those were performed on non-ESKD COVID-19. This reviewer only provided a few examples below, but the authors will need to address this systematically beyond these examples.

3a. Page8, line 269-273, the authors state "...higher lymphocyte related gene signatures and lower myeloid-related ones is a favourable prognostic sign...". These findings are actually consistent with many published works on non-ESKD COVID-19 cohort that decreased proportion of lymphoid cells and increase of myeloid cells are associated increase of disease severity (e.g. PMID: 32669297, 33171100).

Author response

We thank the reviewer for highlighting these omissions. We have amended the text to reflect this and added the above references (Results, lines 276-278).

"Our findings are consistent with studies in non-ESKD COVID-19 cohorts that show that a reduction in lymphoid cell proportion and an increase in myeloid cell proportion are associated with more severe disease (e.g. [8,36])"

3b. Page9, line 291-293, the authors found "...severe/critical patients, there was a progressive drop in the proportion of non-classical monocytes..." This finding is indeed consistent with many published works on non-ESKD COVID-19 cohort that reported the decrease of non-classical monocytes is associated with severity of COVID-19. (e.g. PMID: 32514174, 33171100)

Author response

We have amended the text to highlight that our findings mirror that in non-ESKD patients and have added these and other relevant references (Results, lines 295-297).

"This is consistent with previous studies in non-ESKD cohorts showing an association between a decrease in non-classical monocytes and more severe COVID-19 (e.g. [8,30,37,38])."

3c. In the result section "Longitudinal cytokine/chemokine analysis reveals distinct temporal profiles that distinguish disease severity", the authors reported that different plasma proteins exhibited distinct temporal patterns between severe/critical versus mild/moderate ones. The reviewer is wondering do any of these markers are consistent with (or different from) previous published data on similar plasma omic analysis of non-ESKD COVID-19 cohort (e.g. PMID: 33969320, 32717743, etc.)

Author response

We thank the reviewer for this suggestion. Our analysis used a linear mixed model which included a time x disease course interaction term to identify proteins with distinct temporal profiles according to disease course (defined as the peak severity during the course of the illness). Our model used splines to estimate the dynamic changes in protein levels over time. We again compared our results to that of Filbin *et al.* (PMID: 33969320); in this study, samples were obtained from patients who presented to an emergency department with COVID-19 at 0, 3, 7 and 28 days. The authors employed a similar linear mixed modelling strategy, in which they investigated the interaction between severity and time. There was, however, a subtle

difference in their approach as they treated each time as a categorical variable whereas we analysed time as a continuous variable.

Of the 50 cytokines/cytokine receptors with a significant time x severity interaction term in our study, 32 were also measured using Olink in the study by Filbin *et al.* Of these 32 proteins, they found that 28 had a significant time x severity interaction term. Additionally, Filbin *et al.* also measured protein abundance using the SomaScan platform v4, which measured 46/50 of the cytokines identified in our data (we used SomaScan v4.1). We re-analysed these data using linear mixed models, finding that 33 of the cytokines/cytokine receptors had a significant time x severity interaction term. Thus, we observed a high degree of concordance in terms of plasma cytokines and receptors that have a different longitudinal profile between mild and severe patients in our ESKD cohort and a non-ESKD patient group. This suggests that severe COVID-19 is characterised by common dynamic cytokine changes across patient groups. However, one protein that exhibited a distinct temporal trajectory between severe/critical and mild/moderate cases in our ESKD cohort but no difference in the data of Filbin *et al.* was EPOR (erythropoietin receptor) (new **Supplementary Figure 16**). EPO is produced by the kidneys and ESKD patients are deficient in it. Thus, this appears to be a biologically plausible difference between ESKD and non-ESKD COVID-19 patients.

Changes to the manuscript:

-Supplementary Figure 16 added.

-Lines 337-349.

R2 point 4. *Opportunity to synergize plasma proteomics with PBMC immune cell transcriptome. The authors have performed a lot of analysis separately on plasma proteomics with PBMC immune cell transcriptome, but it will be important to synergize the two, otherwise the value of measuring both in the same cohort will be compromised. The authors did try to do the machine learning with the two datasets together, but the conclusion that the prediction proteome along works even greater than the multi-omic data is not an appropriate stopping point and can be misleading. Is there any other analysis that the authors can do to sell more on the synergy of the two datasets? For example, does the combination of two datasets help separate the patients with different severity better than only using one dataset (e.g. PMID: 34489601, Lee et al showed that combining plasma omic and PBMC transcriptome can classify patients with different severity even better than just plasma data.). Or, does the enrichment results of the plasma omics independently matches the results of immune transcriptomic and thus enhance the message of what biological process are going on in those patients? (e.g. the monocyte transcriptomic and plasma proteomic integration analysis in PMID: 33171100).*

Author response

We disagree with the reviewer's assessment that we have not sufficiently utilised the combination of the PBMC transcriptome and the plasma proteome. We analysed the PBMC transcriptome and the plasma proteome both as separate data layers and also in combination. In addition to the supervised learning analyses (where we compared the performance of the PBMC transcriptome, the plasma proteome and both together in correctly identifying clinical severity), we also performed an integrative multi-omic analysis with MEFISTO (**Supplementary Figure 10**), an extension to Multi-Omics Factor Analysis (MOFA) that identifies factors that change over time (from symptom onset).

While the specific methods used differ, our MEFISTO analysis led to a similar conclusion to that found by the study by Su *et al* referenced by the reviewer (PMID 33171100). Both our

MEFISTO analysis and the integrative network analysis by Su *et al.* identified a single factor in the data that was highly related to COVID-19 severity and pro-inflammatory cytokines. We additionally found that this factor was relatively stable in milder COVID-19, but showed a dynamic temporal profile in more severe disease, rising following symptom onset (**Supplementary Figure 10B**).

However, we gained more meaningful biological insight from WGCNA analyses that analysed the PBMC transcriptome and the plasma proteome as separate data layers, as this provided more granular resolution of the different transcriptomic and proteomic modules that are important in the host response in severe COVID-19. We also reported how the transcriptomic and proteomic modules correlate with one another (**Figure 4C**, Results line 235).

The study of Lee *et al.* (PMID 34489601) which the reviewer refers conducted immune cell transcriptomics and plasma metabolomics using mass spectrometry, rather than proteomics. While Lee *et al.* report that the combination of transcriptome and metabolome provided superior classification of severity, this is distinct and not comparable to the analysis we present (i.e. using measurement of 6,323 proteins using the aptamer-based SomaScan). Thus, it is likely that the difference between the study of Lee *et al.* and our findings relate to what was measured (i.e. MS-based metabolomics vs aptamer-based proteomics).

The reviewer asks whether the combination of data layers improves classification of patient severity. Using Monte Carlo cross-validation, we showed that the proteome provides a better indicator of clinical severity than the PBMC transcriptome (as assessed by AUC), and that combining the two does not improve performance compared to using the plasma proteome alone (**Figure 6**). We found consistent findings using both Random Forests and Lasso methods. We disagree with the reviewer that this is misleading. However, we have added an additional analysis where we ensemble 4 individual models, where each model was developed on a single -omic layer (see our response **R1 point 3** and **R2 point 7**).

As the reviewer suggests, we did identify enrichment of certain pathways that were common to both the PBMC transcriptomic and the plasma proteomic modalities. For instance, histones featured prominently in both analyses (which we elaborate on in the Discussion lines 493-507). Other pathways identified by both modalities included interferon response, granulopoiesis, clotting, TCR activation and cell cycle processes.

Summary of changes:

- We now cite the study of Su *et al.* and highlight the similarity of the results of their multi-omic integration and our MEFISTO analysis
- We cite the study of Lee *et al.* and discuss the apparent differences.
- We have further highlighted in the Discussion the common pathways identified in both the plasma proteomic and PBMC transcriptomic analyses.
- Ensembled model: Supplementary Material pages 6-7.

R2 point 5. *Clarification of differential analysis between mild and severe on longitudinal sample where each patients has multiple blood draws taken. It is not clear to this reviewer how do the authors statistically account for multiple sampling of the same individuals, when performing the differential analysis between mild and severe disease, because there could be multiple blood draws taken from the same individual. This should be clarified and justified.*

Author response

As described in the Results and Methods of our original manuscript, we used linear mixed models (LMMs) to account for the statistical non-independence of repeated measurements taken from the same individual. LMMs contain both fixed and random effects; the explanatory variables are represented as fixed effects and the individual as a random effect. LMMs are a widely used, well-established, and principled approach for analysis of serial samples and longitudinal data (see, for example, Laird, N.M. and Ware, J.H., 1982. Random-effects models for longitudinal data. Biometrics, pp.963-974; Verbeke, G., 1997. Linear mixed models for longitudinal data. In Linear mixed models in practice, pp. 63-153. Springer, New York, NY).

To make this clearer to the reader, we have revised the text of the Results to explicitly spell out that LMMs account for non-independence of serial samples:

Revised text (Results):

“In the Wave 1 cohort, differential gene expression analysis between COVID-19 positive (n=179 samples from 51 patients) and negative samples (n=55) (using linear mixed models (LMM) to account for repeated samples from the same individuals) identified 3,026 significantly up-regulated and 3,329 down-regulated genes (1% false discovery rate, FDR) (**Supplementary File 1A**).”

***R2 point 6.** The title of the manuscript specified “severity predictor” which can be misleading because the word “prediction” generally implies measure something at earlier time point to predict future events. It seems like this is not the case in the analysis of LRRC15. The severity seems to be at the exact same time point of the plasma omic measurement. If that’s the case, the authors should better use words “associated marker” or something like that to replace “predictor” in order to avoid the potential misleading interpretation from readers.*

Author response

The reviewer is correct: in the supervised learning analyses we were ‘predicting’ severity at the time of the sample i.e. we were using the word ‘predictor’ in the statistical/machine learning sense rather than the more general/colloquial sense of prediction of future events. We agree that this ambiguity is open to misinterpretation and have revised the title accordingly.

New title: “Multi-omics identify falling plasma LRRC15 as a COVID-19 severity marker and persistent pro-thrombotic signals in convalescence”

We have also rephrased the relevant sections of the text (changes in purple):.

Results

“Plasma LRRC15 as a marker of COVID-19 severity”

“We identify plasma levels of the proposed alternative SARS-CoV-2 receptor, LRRC15, as a strong marker of COVID-19 severity.”

Discussion:

“A striking finding was the utility of plasma levels of LRRC15 as a marker of COVID-19 severity.”

“We identify plasma levels of the proposed alternative SARS-CoV-2 receptor, LRRC15, as a strong marker of COVID-19 severity.

R2 point 7. *Validation cohorts for machine learning model. A validation cohort is generally needed for machine learning related biomarker analysis. This manuscript already have two cohorts, so why not use one as a training set and the other as an independent validation cohort. Otherwise, it may seems to be using the same dataset to train and “predict” which is not as compelling.*

Author response

Assessment of the performance of supervised learning is commonly performed by splitting the data into training and test (validation) sets; while this approach is generally considered the ‘gold-standard’ it relies on large sample sizes and has several limitations. The limitations of the simple test set/validation split are outlined in the Introduction to Statistical Learning by Tibshirani, Hastie and Friedman (Chapter 5, pp 200):

“1) The validation estimate of the test error rate can be highly variable, depending on precisely which observations are included in the training set and which observations are included in the validation set.

2) In the validation approach, only a subset of the observations—those that are included in the training set rather than in the validation set—are used to fit the model. Since statistical methods tend to perform worse when trained on fewer observations, this suggests that the validation set error rate may tend to overestimate the test error rate for the model fit on the entire data set. “

For relatively small datasets (such as the one in our study), the train/test split approach has been shown to perform poorly in both simulations and real data, likely due to greater uncertainty in both the model fitting and the validation step (e.g. see also Singh et al. Impact of train/test sample regimen on performance estimate stability of machine learning in cardiovascular imaging. Sci Rep 2021; 11: 14490, <https://doi.org/10.1038/s41598-021-93651-5>). An alternative approach that generally works better for smaller datasets is cross-validation, which is what we elected to do in our study. Specifically, we randomly split the data, and use 80% of the data for training and 20% for testing. We repeat the procedure many times: each time randomly splitting the data into training and test sets. Thus, for each iteration there is testing on unseen data and we generate an AUC-ROC estimate. We can then pool these many AUC-ROC estimates to generate a more stable estimate of the true AUC-ROC (with mean and confidence intervals). This AUC-ROC estimate from cross-validation will be more representative than that from a single test/train split.

Additionally, by training the models on data from both the Wave 1 and Wave 2 cohorts, the models are forced to learn features that are common to both datasets (i.e. those that are likely to generalise better).

Nevertheless, at the reviewer’s suggestion, we have now performed an additional supervised learning analysis in which we train the model using the Wave 1 cohort and test it in the Wave 2 cohort (Results lines 394-396, Supplementary Material pages 6-7). These models were useful predictors, with AUC scores between 0.70 and 0.96. However, we emphasise that the point estimates of AUC for these models are less reliable compared to our original analysis, which used cross-validation to create a confidence interval for the model’s performance.

R2 point 8. *Is LRRC15 also a biomarker for severity in non-ESKD COVID-19 cohort, perhaps by analyzing published plasma omic dataset for non-ESKD COVID-19 patient cohort?*

We thank the reviewer for this helpful suggestion. While many of the studies that performed plasma proteomics used assay platforms that did not include LRRC15, there are a few studies that did. We have mined their data to identify whether they identify similar associations with LRRC15.

Su *et al.* MedRxiv 2021 (<https://doi.org/10.1101/2021.10.04.21264015>) performed proteomic profiling with SomaScan in two general population cohorts. Their data indicates that LRRC15 levels are significantly associated with COVID19 outcome (Benjamini-Hochberg adjusted P-value = 10^{-10}), with lower levels associated with severe disease.

A study by Filbin *et al.* (Cell Rep Med 2021; 2: 100287, PMID: 33969320) found that individuals with more severe COVID-19 had a different longitudinal profile of plasma LRRC15 (again measured using SomaScan) compared to less severe cases. As we observed in our data, the abundance of LRRC15 fell over time in more severe patients relative to milder cases (new **Supplementary Figure 22**).

These findings suggest that LRRC15 is also a biomarker of severe COVID-19 in more general patient cohorts.

We have added these observations to the text (lines 423-426, Supplementary Material page 6).

R2 point 9. *Some of the figures may need more clarifications. For example, in Figure 1A, this reviewer is confused regarding what does “+ve” mean for the Wave1 cohort, why Wave 2 cohort does not have “+ve”?*

Author Response

COVID-19 “+ve” is an abbreviation for COVID-19 positive and similarly COVID-19 “-ve” means COVID-19 negative. We have now spelt out ‘positive’ and ‘negative’ to avoid any ambiguity.

With regard to Wave 2, as illustrated on the diagram, this group of 17 patients were initially sampled as COVID-19 negative controls during Wave 1 (2020) but later became infected during Wave 2 in 2021. The 17 patients thus become “COVID-19 +ve” when they are infected. Consequently, for Wave 2 we have pre-infection samples as well as acute infection and convalescent samples from the same individuals. We have expanded the legend to make this clearer.

REVIEWERS' COMMENTS

Reviewer #2 (Remarks to the Author):

No further queries.

Reviewer #3 (Remarks to the Author):

I was asked by the editorial office to evaluate whether the Reviewer #1's comments have been addressed in the revised manuscript. I have intensively reviewed the manuscript, the reviewer comments, and in particular the authors' responses to the comments. All comments of reviewer #1 were answered unreservedly and convincingly by the authors. Only the comment about an evaluation of the study results within the framework of an independent study could not be very convincingly addressed by the authors. This point is certainly very significant, but the lack of a validation cohort should not lead to the rejection of the overall well-founded study.

Due to the extensive revision, the manuscript could be recommended for publication.